# GLAD: Bidirectional Structure-Attribute Alignment via Latent Graph Diffusion Models

Jiankai Zuo [1]   Yu Zhang [2]   Yang Zhang [3]   Zihao Yao [4]   Yaying Zhang[⊠ 4]

## Abstract

Learning on graphs with missing node attributes is a prevalent yet challenging problem in real-world scenarios, as graph neural networks (GNNs) typically rely on complete attribute information. Existing solutions often employ adversarial learning in a shared latent space to align graph structure and attributes. However, these methods frequently suffer from training instability and mode collapse, failing to fully capture the complex, multi-modal joint distribution of topology and features. To address these limitations, we present GLAD, a novel generative framework for robust node attribute completion. GLAD leverages the strong generative capabilities of diffusion models to learn the conditional distribution of attributes given the graph structure within a decoupled latent space. Unlike previous unidirectional approaches, GLAD introduces a robust bidirectional alignment mechanism. Specifically, we incorporate a structure reconstruction constraint during training and structure-aware classifier-free guidance during sampling, ensuring that generated attributes are not only plausible but also maintain strict topological consistency with the underlying graph. Theoretically, we show that GLAD maximizes a tighter variational lower bound on the joint log-likelihood compared to GAN-based predecessors, leading to superior mode coverage. Extensive experiments on large-scale benchmarks demonstrate that GLAD significantly outperforms state-of-the-art baselines in both attribute recovery quality and downstream task performance.

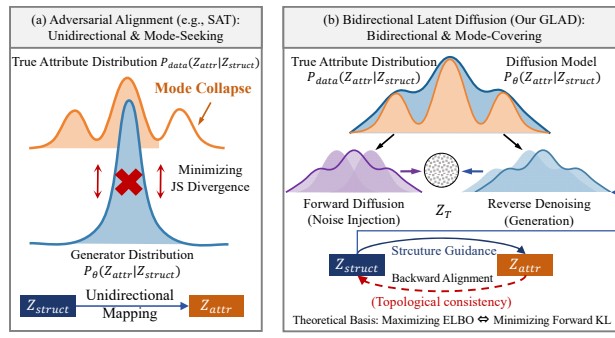

*Figure 1.* A conceptual comparison of attribute completion.

[1]School of Electronic and Information Engineering, Suzhou University of Science and Technology [2]School of Computing, Faculty of Science and Engineering, Macquarie University [3]The Anuradha and Vikas Sinha Department of Data Science, University of North Texas [4]School of Computer Science and Technology, Tongji University. Correspondence to: Yaying Zhang <yaying.zhang@tongji.edu.cn>.

*Proceedings of the 43rd International Conference on Machine Learning*, Seoul, South Korea. PMLR 306, 2026. Copyright 2026 by the author(s).

## 1. Introduction

Graphs are ubiquitous data structures for modeling complex systems characterized by interacting entities, from social networks and biological interactomes to knowledge graphs and recommendation systems (Wu et al., 2020; Sharma et al., 2024). The remarkable success of Graph Neural Networks (GNNs) (Kipf, 2016; Veličković et al., 2017; Corso et al., 2024) in various downstream tasks typically relies on the availability of complete information, where each node is associated with a rich set of attributes alongside the topological structure. However, in many real-world scenarios, node attributes are often partially or entirely missing due to privacy concerns, data collection costs, or noise in the recording process (Zhou et al., 2020; Khemani et al., 2024). This prevalent problem of *attribute-missing graphs* significantly hampers the performance of GNNs designed for attribute-complete data, highlighting the critical need for effective node attribute completion techniques.

Traditional imputation methods (Gama et al., 2025; Zhang et al., 2025), such as mean-filling or simple structural propagation (e.g., label propagation), often fail to capture the complex, non-linear dependencies between graph structure and high-dimensional node features. More recent approaches (Jin et al., 2023; Li et al., 2024; He et al., 2024) have attempted to model the joint distribution of structure and attributes. A prominent line of work employs generative adversarial networks (GANs) to align the distributions of structure and attribute embeddings in a shared latent space, exemplified by the Structure-Attribute Transformer (SAT) (Chen et al., 2022). As shown in Figure 1, while

SAT represents a significant step forward, it inherits the well-known limitations of adversarial training. First, GANs are notoriously unstable to train and prone to *mode collapse*, where the generator produces a limited variety of samples, failing to capture the multi-modal nature of the attribute distribution (i.e., multiple plausible attribute configurations for a given structural context). Second, these methods often rely on a unidirectional alignment assumption (using structure to generate attributes), potentially neglecting the mutual information flow that ensures the completed attributes are fully consistent with the underlying graph topology.

In this paper, we propose to overcome these limitations by harnessing the power of Diffusion Models (Ho et al., 2020; Song et al., 2020b), which have recently emerged as a powerful paradigm for high-fidelity generation. We present GLAD (**G**raph **L**atent **A**ttribute **D**iffusion with Bidirectional Alignment), a novel generative framework specifically designed for robust node attribute completion. GLAD operates in a decoupled latent space to handle the heterogeneity of topological and attribute data. Unlike previous approaches, GLAD models the conditional distribution of missing attributes through a structure-conditioned latent diffusion process, offering stable training and superior mode coverage.

Our key innovation is the introduction of a robust bidirectional alignment mechanism to ensure strict topological consistency. We achieve this through two synergistic components: (1) a Backward Topological Alignment loss during training, which explicitly forces the generated attributes to be capable of reconstructing the original graph structure; and (2) a structure-aware classifier-free guidance (CFG) strategy during sampling (Ho & Salimans, 2022), which allows for controllable generation that strongly favors structurally relevant attribute configurations. Theoretically, we show that GLAD's training objective is equivalent to maximizing a tighter variational lower bound (ELBO) on the joint log-likelihood compared to adversarial baselines, providing a formal justification for its superior ability to capture diverse attribute patterns.

Our main contributions are summarized as follows:

- We propose GLAD, a novel generative framework that leverages latent diffusion models for robust node attribute completion on graphs, addressing the instability and mode collapse issues of prior GAN-based methods.

- We introduce a bidirectional alignment mechanism, combining a backward topological reconstruction constraint and structure-aware classifier-free guidance, to ensure strong consistency between generated attributes and graph topology.

- We provide a theoretical analysis demonstrating that GLAD optimizes a tighter variational lower bound

(ELBO) on the joint distribution, leading to superior mode coverage compared to adversarial approaches.

- Extensive evaluations show that GLAD achieves state-of-the-art performance on attribute completion and various downstream tasks across multiple benchmarks.

## 2. Related Work

### 2.1. Graph Learning with Missing Attributes

Handling missing data in graph learning is a pivotal challenge. Early approaches relied on heuristic imputation methods such as mean filling or zero padding, which often ignore the correlation between node features and graph topology. Subsequently, propagation-based methods like GINN (Spinelli et al., 2020) were proposed to diffuse observed feature information across edges. While effective for homophilic graphs, it struggles to model complex, nonlinear feature-structure dependencies. More recently, some models have been adopted to learn the joint distribution of features and structure. For example, the Structure-Attribute Transformer (SAT) (Chen et al., 2022) employs a GAN-based architecture to align decoupled structure and attribute representations in a shared latent space. Despite achieving excellent results, SAT relies on adversarial training, which is notoriously unstable and prone to mode collapse, limiting its ability to capture the multi-modal nature of missing attributes. In contrast, GLAD replaces the adversarial alignment with a stable, likelihood-based latent diffusion process, ensuring superior mode coverage and training stability.

### 2.2. Deep Generative Models on Graphs

Generative models for graph data have evolved significantly. Early works like GraphVAE (Simonovsky & Komodakis, 2018) focused primarily on generating graph topology. Recent advances have shifted towards generating both structure and attributes. For instance, Amer (Jin et al., 2023) autoregressively generates nodes and edges, CSAT (Li et al., 2024) employs a contrastive sampling, WAGE (Tu et al., 2025) proposes a weight-sharing graph autoencoder, and WGNN (Chen et al., 2025) uses a Wasserstein-based GNN. However, they are typically designed for *de novo* graph generation rather than the conditional completion of missing attributes in a fixed graph structure. Applying them to the attribute completion is non-trivial, as it requires conditioning the generation process on the existing partial observations and the specific graph topology, a challenge that GLAD explicitly addresses through its structure-conditioned design.

### 2.3. Diffusion Probabilistic Models

Diffusion models (Yang et al., 2023; Cao et al., 2024) have demonstrated unprecedented success in computer vision

and natural language processing. Recently, diffusion models have been adapted to the graph domain. GDSS (Jo et al., 2022) defines a system of stochastic differential equations (SDEs) to generate nodes and edges jointly in continuous space. DiGress (Vignac et al., 2022) introduces discrete diffusion processes for categorical node attributes and edges. However, existing graph diffusion models primarily focus on the unconditional generation of small-scale molecules or generic graphs. They do not directly address the *attribute imputation* problem, where the goal is to recover missing values conditioned on observed data and topology. Furthermore, applying diffusion directly in the high-dimensional node attribute space is computationally expensive and difficult to scale. **GLAD** bridges this gap by introducing the first *latent* diffusion framework tailored for graph attribute completion. By performing diffusion in a compressed, decoupled latent space and introducing a novel bidirectional alignment mechanism, GLAD achieves both computational efficiency and high-fidelity conditional generation.

## 3. Method

In this section, we present the GLAD (**G**raph **L**atent **A**ttribute **D**iffusion), a generative framework designed for robust node attribute completion. As illustrated in Figure 2, GLAD operates in a decoupled latent space and employs a novel bidirectional alignment mechanism via structure-conditioned diffusion to model the complex joint distribution of graph topology and attributes.

### 3.1. Problem Formulation

Let $\mathcal{G} = (\mathcal{V}, \mathcal{E}, \mathbf{A}, \mathbf{X})$ denote an attribute-missing graph, where $\mathcal{V}$ is the set of $N$ nodes, $\mathcal{E}$ is the edge set, and $\mathbf{A} \in \{0,1\}^{N \times N}$ is the adjacency matrix. The attribute matrix $\mathbf{X} \in \mathbb{R}^{N \times F}$ is *partially observed*. Let $\mathcal{V}_{obs}$ and $\mathcal{V}_{miss}$ denote the sets of nodes with observed and missing attributes, respectively, such that $\mathcal{V}_{obs} \cup \mathcal{V}_{miss} = \mathcal{V}$. The observed attribute matrix is denoted as $\mathbf{X}_{obs}$, containing rows indexed by $\mathcal{V}_{obs}$, while rows indexed by $\mathcal{V}_{miss}$ are unknown. Our goal is to learn a conditional generative model $p_\theta(\mathbf{X}_{miss}|\mathbf{A}, \mathbf{X}_{obs})$ to impute the missing attributes, ensuring the generated attributes are topologically consistent with the graph structure.

### 3.2. Decoupled Latent Encoding (Phase 1)

To handle the heterogeneity between high-dimensional graph topology and diverse node attributes, we first project them into decoupled low-dimensional latent spaces, following the shared-latent space assumption (Chen et al., 2022).

As shown in Figure 2 (Phase 1), we employ a Graph Neural Network (GNN) as the structure encoder $E_{struct}$ to capture topological information: $\mathbf{Z}_{struct} = E_{struct}(\mathbf{A}) \in \mathbb{R}^{N \times d_s}$.

Simultaneously, for observed nodes, an MLP-based attribute encoder $E_{attr}$ maps attributes to attribute latents: $\mathbf{Z}_{attr} = E_{attr}(\mathbf{X}_{obs}) \in \mathbb{R}^{|\mathcal{V}_{obs}| \times d_a}$. The missing parts of $\mathbf{Z}_{attr}$ corresponding to $\mathcal{V}_{miss}$ are initialized with Gaussian noise, ready to be refined by the subsequent diffusion process.

### 3.3. Structure-Conditioned Latent Diffusion (Phase 2)

The core of GLAD is modeling the conditional distribution of attribute latents given structure latents, $p(\mathbf{Z}_{attr}|\mathbf{Z}_{struct})$, using a Latent Diffusion Model (LDM). This avoids the instability of adversarial matching (Xing et al., 2021).

**Forward Diffusion Process.** We define a fixed Markovian forward process that gradually adds Gaussian noise to the attribute latents $\mathbf{Z}_{attr}$ (denoted as $\mathbf{z}_0$ for simplicity in this subsection) over $T$ steps. The distribution of $\mathbf{z}_t$ at timestep $t$ given $\mathbf{z}_{t-1}$ is:

$$q(\mathbf{z}_t|\mathbf{z}_{t-1}) = \mathcal{N}(\mathbf{z}_t; \sqrt{1-\beta_t}\mathbf{z}_{t-1}, \beta_t\mathbf{I}), \quad (1)$$

where $\{\beta_t\}_{t=1}^T$ is a fixed variance schedule. As $T \to \infty$, $\mathbf{z}_T$ approaches a standard Gaussian prior $\mathcal{N}(\mathbf{0}, \mathbf{I})$.

**Reverse Denoising Process.** The generative process learns to reverse this noise injection, conditioned on the structural information $\mathbf{Z}_{struct}$. We parameterize the reverse process as:

$$p_\theta(\mathbf{z}_{t-1}|\mathbf{z}_t, \mathbf{Z}_{struct}) = \mathcal{N}(\mathbf{z}_{t-1}; \boldsymbol{m}_\theta, \boldsymbol{\Sigma}_\theta(\mathbf{z}_t, t)). \quad (2)$$

where $\boldsymbol{m}_\theta = \boldsymbol{\mu}_\theta(\mathbf{z}_t, t, \mathbf{Z}_{struct})$. To estimate $\boldsymbol{\mu}_\theta$, we train a *time-conditional denoising network* $\boldsymbol{\epsilon}_\theta(\mathbf{z}_t, t, \mathbf{Z}_{struct})$ to predict the noise added at each step. Crucially, as depicted in the central module of Figure 2, the structural structure latents $\mathbf{Z}_{struct}$ are injected into the denoising network via multi-head cross-attention layers, serving as the condition to guide the attribute generation process.

The training objective for the diffusion component is a reweighted variation of the evidence lower bound (ELBO), simplified to mean squared error loss on noise prediction for observed nodes:

$$\mathcal{L}_{diff} = \mathbb{E}_{t \sim [1,T], \boldsymbol{\epsilon} \sim \mathcal{N}(\mathbf{0}, \mathbf{I})} \left[ \|\boldsymbol{\epsilon} - \boldsymbol{\epsilon}_\theta(\mathbf{z}_t, t, \mathbf{Z}_{struct})\|^2 \right],$$
$$(3)$$

where $\mathbf{z}_t = \sqrt{\bar{\alpha}_t}\mathbf{z}_0 + \sqrt{1-\bar{\alpha}_t}\boldsymbol{\epsilon}$ ($\mathbf{z}_0 \sim \mathbf{Z}_{attr}^{obs}$), and $\bar{\alpha}_t = \prod_{i=1}^t (1-\beta_i)$.

### 3.4. Bidirectional Alignment and Guidance (Phase 3)

Existing methods (Chen et al., 2022; Jin et al., 2023; Li et al., 2024; Tu et al., 2025) typically rely on unidirectional alignment (Structure → Attribute). A key innovation of GLAD is the introduction of a bidirectional alignment mechanism, ensuring that generated attributes are not only conditioned on the structure but also contain sufficient information to reconstruct the topology.

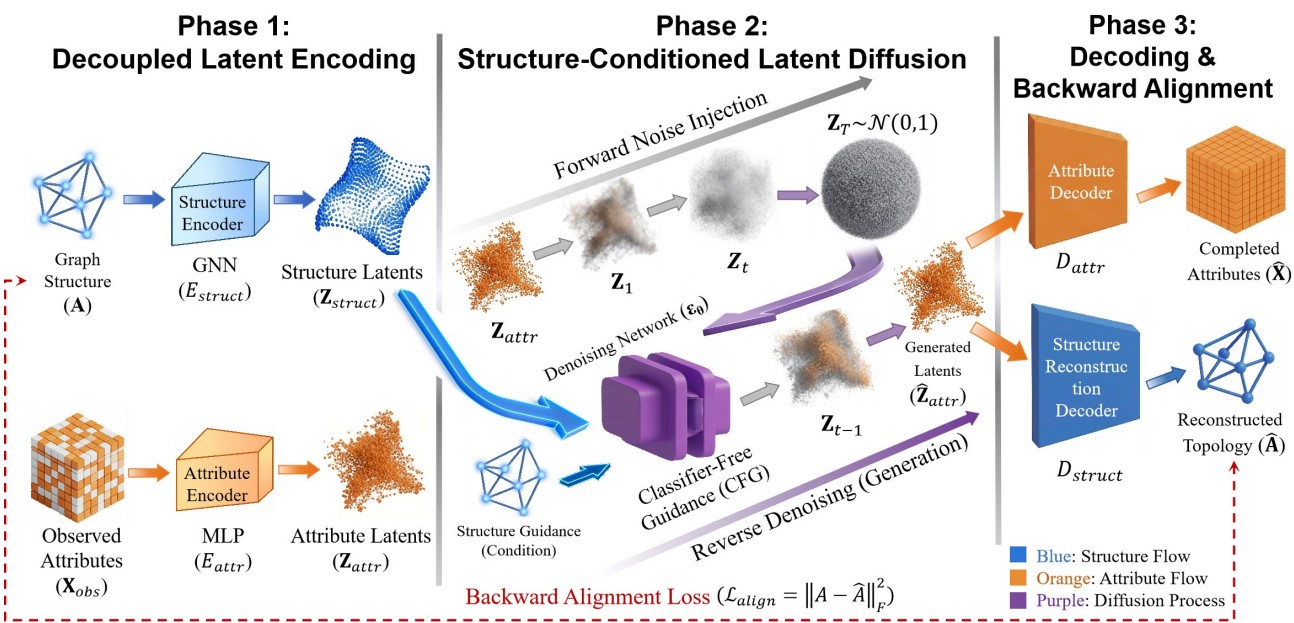

*Figure 2.* The overall architecture of the proposed Graph Latent Attribute Diffusion with Bidirectional Alignment (GLAD) framework.

### 3.4.1. STRUCTURE-AWARE CLASSIFIER-FREE GUIDANCE (SAMPLING)

To enhance the structural consistency of generated attributes during inference, we adapt a Classifier-Free Guidance (CFG) to graph data. During training, we randomly drop the condition $\mathbf{Z}_{struct}$ with probability $p_{uncond}$ to train both a *conditional noise estimate* $\boldsymbol{\epsilon}_\theta(\mathbf{z}_t, \mathbf{Z}_{struct})$ and an *unconditional estimate* $\boldsymbol{\epsilon}_\theta(\mathbf{z}_t, \emptyset)$. During sampling, the guided noise estimate $\tilde{\boldsymbol{\epsilon}}_\theta$ is formulated as:

$$\tilde{\boldsymbol{\epsilon}}_\theta(\mathbf{z}_t, \mathbf{Z}_{struct}) = (1+w)\boldsymbol{\epsilon}_\theta(\mathbf{z}_t, \mathbf{Z}_{struct}) - w\boldsymbol{\epsilon}_\theta(\mathbf{z}_t, \emptyset), \quad (4)$$

where $w \geq 0$ is the guidance scale. A larger $w$ forces the generative process toward regions with higher structural relevance, which is particularly beneficial for homophilic graphs where attributes and structure are highly correlated.

### 3.4.2. BACKWARD TOPOLOGICAL ALIGNMENT CONSTRAINT (TRAINING)

To enforce bidirectional consistency explicitly during training, we introduce a Backward Topological Alignment loss (indicated by the red dashed line in Figure 2, Phase 3).

Latents generated by the diffusion process at $t = 0$, denoted as $\hat{\mathbf{Z}}_{attr}$, are first decoded back to the attribute space by $D_{attr}$, yielding $\hat{\mathbf{X}} = D_{attr}(\hat{\mathbf{Z}}_{attr})$. Simultaneously, we employ a structure reconstruction decoder $D_{struct}$ (e.g., an inner-product decoder) to predict the graph topology from these generated attribute features: $\hat{\mathbf{A}} = D_{struct}(\hat{\mathbf{Z}}_{attr})$. The backward alignment loss minimizes the reconstruction error of the adjacency matrix:

$$\mathcal{L}_{align} = \|\mathbf{A} - \hat{\mathbf{A}}\|_F^2. \quad (5)$$

Alternatively, $\mathcal{L}_{align} = -\frac{1}{|\mathcal{E}|}\sum_{(i,j)\in\mathcal{E}} \log(\hat{A}_{ij}) - \frac{1}{|\mathcal{V}|^2-|\mathcal{E}|}\sum_{(i,j)\notin\mathcal{E}} \log(1 - \hat{A}_{ij})$ (for sparse graphs). This constraint ensures that the generated attributes $\hat{\mathbf{X}}$ preserve the underlying topological community structure, maximizing the mutual information $I(\mathbf{A}; \mathbf{X})$ implicitly.

### 3.5. Overall Training Objective

The final training objective of GLAD is a weighted sum of the diffusion loss and the backward alignment loss:

$$\mathcal{L}_{total} = \mathcal{L}_{diff} + \lambda \cdot \mathcal{L}_{align}, \quad (6)$$

where $\lambda$ balances the generative quality and topological consistency. Appendix D provides the training process.

### 3.6. Theoretical Analysis

Previous approaches like SAT utilize adversarial training to align latent distributions of structure and attributes. We provide a theoretical justification for why the diffusion-based approach in GLAD is superior for the attribute completion task, focusing on the training objective and mode coverage.

Let $p_{data}(\mathbf{X}|\mathbf{A})$ denote the true conditional distribution of attributes given structure, and $p_\theta(\mathbf{X}|\mathbf{A})$ be the distribution modeled by GLAD. See Appendix E.1 for detailed proofs.

**Proposition 3.1** (Training Objective Comparison). *Adversarial alignment methods implicitly minimize the Jensen-Shannon (JS) divergence between the latent distributions of structure and attributes. In contrast, optimizing the diffusion loss $\mathcal{L}_{diff}$ in GLAD is equivalent to maximizing the Evidence Lower Bound (ELBO) of the*

*data log-likelihood, which effectively minimizes an upper bound of the forward Kullback-Leibler (KL) divergence, $D_{KL}(p_{data}(\mathbf{X}|\mathbf{A})\|p_\theta(\mathbf{X}|\mathbf{A}))$.*

Based on Proposition 3.1, we derive important implications regarding the quality of generated attributes, specifically concerning mode coverage (i.e., the ability to capture diverse attribute patterns corresponding to a given structure).

**Theorem 3.2** (Superior Mode Coverage of GLAD). *Let $\mathcal{M}_{data} = \{\mathbf{X} : p_{data}(\mathbf{X}|\mathbf{A}) > 0\}$ be the support of the true conditional distribution (i.e., all plausible attribute configurations for graph $\mathbf{A}$). The distribution $p_\theta^*$ learned by minimizing the forward KL divergence in GLAD tends to cover the entire support $\mathcal{M}_{data}$. Conversely, distributions learned by minimizing JS divergence (adversarial methods) are prone to mode collapse, where the learned support forms a subset of the true support.*

Theorem 3.2 suggests that GLAD is theoretically better suited for graph attribute completion, especially in complex graphs where multiple plausible attribute configurations may exist for the same structural context (high uncertainty). GLAD is designed to capture this multi-modal uncertainty, whereas GAN-based baselines tend to produce singular, deterministic completions. Detailed proofs in Appendix E.2.

**Theorem 3.3** (Information Consistency via Mutual Information). *Let $\mathcal{L}_{align} = \|\mathbf{A} - \hat{\mathbf{A}}\|^2$ be the backward topological alignment loss, where $\hat{\mathbf{A}} = D_{struct}(\mathbf{Z}_{attr})$. Minimizing $\mathcal{L}_{align}$ is equivalent to maximizing a variational lower bound on the mutual information $I(\mathbf{A}; \mathbf{Z}_{attr})$ between the graph topology $\mathbf{A}$ and the generated attribute latents $\mathbf{Z}_{attr}$.*

Under the assumption of a Gaussian observation model for the structure decoder (or Bernoulli for discrete edges), the negative log-likelihood $-\log q_\phi(\mathbf{A}|\mathbf{Z}_{attr})$ simplifies to the mean squared error (or cross-entropy) loss $\mathcal{L}_{align}$. Thus, minimizing $\mathcal{L}_{align}$ tightens the variational lower bound of $I(\mathbf{A}; \mathbf{Z}_{attr})$, ensuring that the generated attributes encapsulate sufficient structural information.

Theorem 3.3 provides an information-theoretic justification for our bidirectional alignment mechanism. While traditional GAN-based methods (e.g., SAT) only focus on the mapping $\mathbf{A} \rightarrow \mathbf{X}$, GLAD explicitly reinforces the reverse dependency $\mathbf{X} \rightarrow \mathbf{A}$. By maximizing the mutual information, we ensure that the generated attributes are not merely "noise-free" versions of features, but are fundamentally "anchored" to the graph's community structure and local connectivity. The complete proof is detailed in Appendix E.3.

### 3.7. Complexity Analysis

In this section, we analyze the computational complexity of GLAD in terms of both time and space. Let $N$ be the number of nodes, $|\mathcal{E}|$ the number of edges, $d$ the dimension of the latent space, and $T$ the number of diffusion steps.

**Time Complexity.** The total time complexity of GLAD consists of three components: i) Encoding Phase: The structure encoder (GNN) typically takes $O(|\mathcal{E}| \times d + N \times d^2)$, and the attribute encoder (MLP) takes $O(N \times d^2)$. ii) Diffusion Phase (Training): Each training step involves a forward pass through the denoising network $\epsilon_\theta$. By employing a cross-attention mechanism between node-wise structure and attribute latents, the complexity per step is $O(N \times d^2)$, which is linear with respect to the number of nodes. Unlike graph-wide generation models that scale with $O(N^2)$, GLAD's node-wise latent diffusion is highly efficient. iii) Bidirectional Alignment: The backward topological alignment loss involves a structure decoder. By using negative sampling, we can reduce the complexity of reconstructing the adjacency matrix from $O(N^2)$ to $O(|\mathcal{E}| \times d)$, making it scalable to sparse real-world graphs.

**Inference Cost.** During sampling, GLAD requires $T$ iterations of the reverse denoising process. Although the inference time is $O(T \times N \times d^2)$, the use of the *latent space* (where $d \ll F$, the original feature dimension) significantly reduces the constant factor compared to feature-space diffusion models. Efficiency can be further improved using fast samplers like DDIM (Song et al., 2020a) to reduce $T$.

**Space Complexity.** GLAD stores the latent representations and the graph structure. The space complexity is $O(N \times d + |\mathcal{E}|)$, which is consistent with GNN-based models. Since the diffusion process occurs in the latent space, the memory footprint for the denoising network is independent of the original feature dimension $F$, allowing GLAD to handle datasets with extremely high-dimensional attributes.

## 4. Experiments

In this section, we rigorously evaluate GLAD to answer the following research questions:

- **Q1 (Attribute Recovery Quality):** Can GLAD recover missing node attributes more accurately than GAN-based and propagation-based baselines?

- **Q2 (Downstream Utility):** Does the enhanced attribute quality translate into superior performance in standard downstream tasks like node classification?

- **Q3 (Ablation Study):** What is the contribution of the bidirectional alignment and structure-aware guidance components?

- **Q4 (Real-World Generalization):** How does GLAD perform on complex, real-world sensor networks (e.g., traffic systems) with continuous and noisy data?

*Table 1.* Profiling of the attribute-level evaluation for node attribute reconstruction on the Cora and Citeseer dataset.

| Method | Cora | | | | | | Citeseer | | | | | |
|--------|------|------|------|------|------|------|------|------|------|------|------|------|
| | R@10 | R@20 | R@50 | N@10 | N@20 | N@50 | R@10 | R@20 | R@50 | N@10 | N@20 | N@50 |
| VAE | 8.87 | 12.33 | 20.97 | 12.23 | 14.56 | 19.16 | 3.82 | 6.69 | 12.94 | 6.01 | 8.40 | 12.50 |
| GCN | 12.56 | 17.85 | 29.73 | 17.21 | 20.76 | 27.04 | 6.28 | 10.97 | 20.49 | 10.31 | 14.21 | 20.44 |
| GAT | 12.67 | 17.93 | 29.70 | 17.30 | 20.87 | 27.10 | 5.62 | 10.12 | 19.56 | 8.79 | 12.53 | 18.71 |
| NEIGHAGGR | 9.06 | 14.13 | 19.61 | 12.17 | 15.48 | 18.50 | 5.11 | 9.08 | 15.01 | 8.23 | 11.55 | 15.60 |
| GRAPHSAGE | 12.91 | 18.10 | 30.24 | 17.97 | 21.45 | 27.86 | 5.60 | 10.63 | 19.90 | 9.78 | 13.56 | 19.97 |
| GRAPHMAE | 3.69 | 4.39 | 7.48 | 5.74 | 7.25 | 10.79 | 1.27 | 3.54 | 8.35 | 2.23 | 4.91 | 7.51 |
| ARWMF | 12.99 | 18.03 | 29.80 | 18.64 | 22.14 | 27.96 | 5.56 | 10.18 | 19.59 | 8.46 | 12.25 | 18.29 |
| T2-GNN | 12.26 | 15.35 | 22.15 | 17.20 | 19.28 | 22.81 | 5.27 | 8.59 | 15.54 | 9.46 | 12.24 | 16.79 |
| GINN | 13.12 | 18.67 | 28.87 | 18.28 | 21.66 | 27.78 | 6.07 | 10.53 | 20.22 | 9.31 | 13.46 | 19.89 |
| SAT | 14.75 | 21.30 | 33.24 | 20.71 | 24.98 | 31.71 | 7.55 | 12.61 | 23.38 | 13.05 | 17.26 | 24.25 |
| AMER | 14.48 | 20.22 | 31.89 | 20.01 | 23.92 | 30.61 | 7.45 | 11.82 | 21.94 | 12.23 | 16.47 | 22.75 |
| WGNN | 15.21 | 22.34 | 33.20 | 21.22 | 24.83 | 31.62 | 8.00 | 12.25 | 22.33 | 13.15 | 17.59 | 24.02 |
| **GLAD** | 17.32 | 24.55 | 37.03 | 23.91 | 28.83 | 36.12 | 9.77 | 15.52 | 26.86 | 17.65 | 20.15 | 27.22 |

*Note:* The highest performance is shaded in green , and the second-highest performance is shaded in blue .

## 4.1. Experimental Setup

**Datasets.** We evaluate our method on two distinct categories of datasets: (1) **Standard Benchmarks:** We use four widely-used graph datasets: Cora and Citeseer (citation networks), and Amazon-Computer and Amazon-Photo (co-purchase networks). (2) **Real-World Traffic Networks:** To test robustness in industrial scenarios, we utilize three urban traffic datasets collected from loop detectors in London, Madrid, and Melbourne. In these graphs, nodes represent sensors, edges represent road connectivity, and attributes correspond to historical traffic speeds and flow readings. More dataset details are included in Appendix A.

**Baselines.** We compare GLAD against three categories of state-of-the-art methods: (1) Attribute-Complete (AC) methods: VAE (Kingma & Welling, 2013), GCN (Kipf, 2016), GAT (Veličković et al., 2017), NEIGHAGGR (Şimşek & Jensen, 2008), GRAPHSAGE (Hamilton et al., 2017), GRAPHMAE (Hou et al., 2022), and ARWMF (Chen et al., 2019). (2) Attribute-Incomplete (AI) methods: T2-GNN (Huo et al., 2023) and GINN (Spinelli et al., 2020). (3) Attribute-Missing (AM) methods: SAT (Chen et al., 2022), AMER (Jin et al., 2023), and WGNN (Chen et al., 2025).

## 4.2. Performance on Standard Benchmarks (Q1 & Q2)

### 4.2.1. PROFILING ATTRIBUTE RECOVERY

We first evaluate the direct quality of the imputed attributes as a reconstruction task. Table 1 summarizes the results on Cora and Citeseer datasets, and Appendix F provides additional experimental results. GLAD consistently outperforms all baselines across all metrics (Recall@k and NDCG@k). Specifically, on the Cora dataset (citation

*Table 2.* Node classification of the node-level evaluation for node attribute completion.

| Type | Method | Cora | Citeseer | Amazon-C | Amazon-P |
|------|--------|------|----------|----------|----------|
| AC | VAE | 29.29 | 26.69 | 44.38 | 52.45 |
| | GCN | 44.27 | 40.52 | 40.19 | 37.77 |
| | GAT | 45.95 | 27.09 | 40.15 | 37.80 |
| | NEIGHAGGR | 64.76 | 54.19 | 87.13 | 90.10 |
| | GRAPHSAGE | 60.13 | 43.67 | 40.13 | 38.26 |
| | GRAPHMAE | 75.06 | 67.89 | 84.37 | 88.26 |
| | ARWMF | 80.65 | 27.23 | 73.88 | 61.78 |
| AI | T2-GNN | 74.49 | 67.33 | 62.35 | 71.29 |
| | GINN | 67.58 | 55.32 | 81.72 | 87.77 |
| AM | SAT-GCN | 83.24 | 66.04 | 84.98 | 89.12 |
| | SAT-GAT | 85.35 | 67.32 | 87.60 | 92.50 |
| | AMER | 80.21 | 66.95 | 79.06 | 90.46 |
| | WGNN | 82.74 | 65.70 | 88.18 | 92.00 |
| Ours | **GLAD** | 86.05 | 69.58 | 88.72 | 92.41 |

graphs), GLAD achieves a Recall@10 of 17.32, surpassing the strongest GAN-based baseline WGNN (15.21) by 13.9%. A similar trend is observed on Citeseer, where GLAD reaches an NDCG@50 of 27.22. These results demonstrate that by leveraging the latent diffusion process and bidirectional alignment, GLAD captures the multimodal distribution of node attributes more effectively than adversarial or propagation-based methods, leading to higher-fidelity attribute reconstruction.

### 4.2.2. NODE CLASSIFICATION ACCURACY

To verify the downstream utility of the completed graphs, we perform node classification using a standard GCN trained on the recovered graph. As shown in Table 2, GLAD achieves state-of-the-art performance across all four benchmarks. Notably, on the Citeseer dataset, GLAD reaches an average

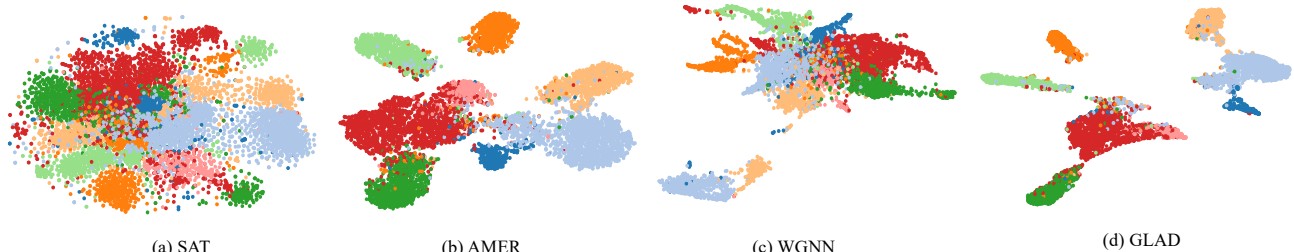

(a) SAT      (b) AMER      (c) WGNN      (d) GLAD

*Figure 3.* Visualization of node representations generated by four methods on the Amazon-Photo dataset.

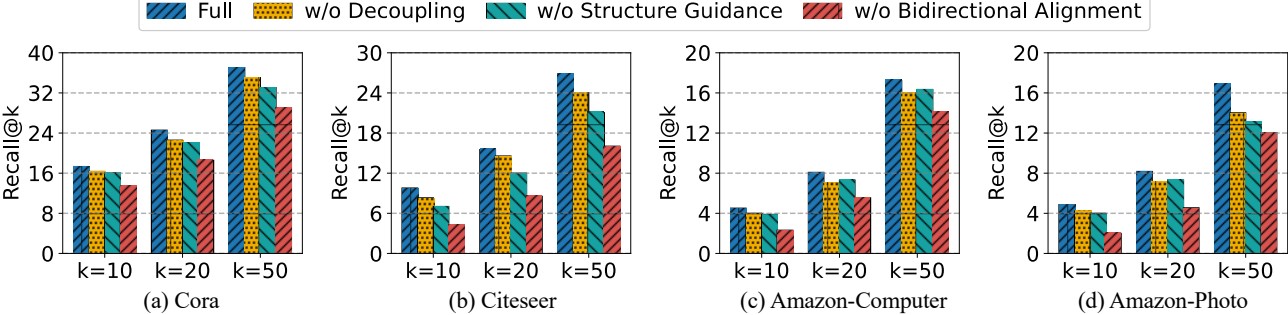

*Figure 4.* Ablation results of the proposed GLAD model across four graph datasets in term of the Recall@k evaluation metric.

accuracy of 69.58%, providing a significant improvement over the best attribute-missing (AM) baseline SAT-GCN (67.32%). On the larger Amazon-Computer and Amazon-Photo datasets, GLAD continues to exhibit superior performance. The consistent gains in classification accuracy indicate that the attributes generated by GLAD are not only statistically plausible but also preserve the semantic information necessary for downstream discriminative tasks.

To qualitatively evaluate the discriminative power of the completed node attributes, we employ t-SNE to visualize the node representations. As illustrated in Figure 3, GLAD exhibits significantly clearer and more compact cluster structures compared to the GAN-based baselines. Specifically, while SAT and AMER suffer from substantial overlap between different classes due to their inability to fully capture the complex joint distribution, GLAD produces well-separated clusters with high intra-class compactness and distinct inter-class boundaries. Although WGNN improves upon previous methods, it still exhibits blurred regions between clusters. The superior clustering quality of GLAD provides a visual explanation for its state-of-the-art classification accuracy, demonstrating that our bidirectional alignment mechanism effectively preserves the underlying semantic structure during the latent diffusion process.

### 4.3. Ablation Study (Q3)

To investigate the contribution of each component in GLAD, we conduct an ablation study with three variants: (1) *w/o*

*Bidirectional Alignment* ($\lambda = 0$): Removes the backward topological reconstruction loss. (2) *w/o Structure Guidance* ($w = 0$): Removes the classifier-free guidance, relying only on the base condition. (3) *w/o Decoupling (Joint Space)*: Performs diffusion directly on the concatenated $A$ and $X$ space, verifying the necessity of our latent architecture.

Figure 4 illustrates the ablation results. We observe that removing the *Bidirectional Alignment* leads to the most substantial performance drop (decreased by nearly 30% in Recall@10), confirming that the backward topological constraint is crucial for ensuring that generated attributes remain consistent with the graph structure. Furthermore, the exclusion of *Structure Guidance* results in a noticeable decrease on the Citeseer, highlighting the effectiveness of our structure-aware classifier-free guidance in steering the diffusion process.

Finally, the *Decoupling* strategy also contributes to the overall stability and quality, as performing diffusion in a joint space tends to suffer from the heterogeneity between topological and attribute data. These results validate that all three components are synergistic and essential for the success of the Graph Latent Attribute Diffusion (GLAD).

### 4.4. Real-world Application: City Traffic Analysis (Q4)

Traffic data is inherently noisy and exhibits complex spatial correlations, serving as a rigorous testbed for model generalization. We conduct two distinct downstream tasks.

*Table 3.* Performance comparison of the estimated time of arrival (ETA) in terms of MAE, RMSE, and MAPE. Lower is better.

| Dataset | Metrics | GCN | GAT | SAT | AMER | WGNN | START | JGRM | DYFFUSION | DIFFSTG | **GLAD** |
|---------|---------|-----|-----|-----|------|------|-------|------|-----------|---------|----------|
| London | MAE | 146.34 | 115.41 | 97.74 | 95.09 | 93.17 | 93.88 | 91.22 | 89.20 | 87.55 | 80.01 |
| | RMSE | 285.58 | 223.05 | 207.67 | 205.40 | 201.03 | 202.94 | 199.03 | 196.32 | 193.65 | 182.52 |
| | MAPE | 32.75 | 24.32 | 21.07 | 20.46 | 19.64 | 19.60 | 19.45 | 19.05 | 17.88 | 14.95 |
| Madrid | MAE | 105.02 | 80.34 | 65.51 | 64.00 | 61.09 | 62.34 | 59.30 | 56.75 | 57.06 | 53.17 |
| | RMSE | 200.45 | 178.93 | 146.97 | 143.85 | 141.82 | 143.28 | 140.88 | 138.22 | 138.14 | 136.75 |
| | MAPE | 25.07 | 20.12 | 16.35 | 15.20 | 14.88 | 16.02 | 14.03 | 13.01 | 13.25 | 11.69 |
| Melbourne | MAE | 65.29 | 60.23 | 45.17 | 42.33 | 39.34 | 41.28 | 40.52 | 38.55 | 36.03 | 34.50 |
| | RMSE | 183.66 | 179.01 | 140.26 | 138.40 | 130.84 | 135.02 | 132.21 | 129.02 | 129.25 | 126.49 |
| | MAPE | 22.05 | 21.05 | 17.14 | 13.23 | 11.92 | 12.99 | 12.58 | 10.30 | 10.35 | 9.25 |

#### 4.4.1. TASK 1: ESTIMATED TIME OF ARRIVAL (ETA)

We aim to predict the exact travel time (a continuous variable) based on the completed traffic speed map. This requires the model to capture fine-grained numerical distributions rather than just classes. We employ Mean Absolute Error (MAE), Root Mean Square Error (RMSE), and Mean Absolute Percentage Error (MAPE).

Beyond the existing GNN and GAN-based models, we incorporate two representation-learning-based baselines: START (Jiang et al., 2023) and JGRM (Ma et al., 2024); two diffusion-based spatio-temporal forecasting methods: DYF-FUSION (Rühling Cachay et al., 2023) and DIFFSTG (Wen et al., 2023). Table 3 reports the results, and GLAD significantly outperforms all baselines. For instance, in the London dataset, GLAD reduces the MAE to 80.01, a 8.6% improvement over DIFFSTG (87.55). The superior performance in RMSE further suggests that GLAD's diffusion-based approach is more robust to outliers and better at capturing the multi-modal uncertainty of traffic states compared to deterministic or GAN-based models (e.g., SAT and WGNN).

#### 4.4.2. TASK 2: TRAFFIC CONGESTION CLASSIFICATION

Finally, we test the model's ability to recover categorical states via congestion level classification (i.e., Free-flow, Slow, Congested). As shown in Table 4, GLAD achieves the highest accuracy across all cities, reaching 87.65% in London and over 93% in Madrid and Melbourne. Compared to START and AMER, GLAD's ability to maintain strict topological consistency through bidirectional alignment allows it to better infer congestion patterns from the surrounding road network structure, even when a large portion of sensor data is missing. This tests the model's ability to recover categorical states from structural context.

### 4.5. Diversity & Coverage Evaluation

To rigorously validate our central claims regarding mode coverage and topological consistency, we have conducted targeted validation experiments under a highly ambiguous

*Table 4.* Performance comparison of traffic classification.

| Method | London | Madrid | Melbourne |
|--------|--------|--------|-----------|
| VAE | 36.22 | 45.28 | 50.02 |
| GCN | 50.33 | 52.78 | 60.44 |
| GAT | 51.77 | 53.19 | 60.68 |
| GRAPHSAGE | 50.65 | 54.20 | 62.39 |
| T2-GNN | 60.09 | 67.66 | 71.59 |
| GINN | 65.57 | 60.12 | 61.92 |
| SAT | 75.44 | 80.20 | 83.21 |
| AMER | 78.11 | 78.56 | 78.07 |
| WGNN | 74.45 | 80.70 | 84.24 |
| START | 78.68 | 77.58 | 86.05 |
| JGRM | 82.55 | 82.34 | 88.15 |
| DYFFUSION | 81.04 | 85.64 | 90.36 |
| DIFFSTG | 80.59 | 90.23 | 88.08 |
| **GLAD** | 87.65 | 93.38 | 94.12 |

setting (Cora dataset with an 80% attribute missing rate). We introduce two direct quantitative metrics:

1. Mode Coverage (Diversity) via Average Pairwise Distance (APD): To explicitly test if adversarial baselines collapse to fewer completions, we generate $K = 10$ different samples for each missing node using varying random seeds/noise. We compute the average Euclidean distance among these $K$ completions. An APD close to 0 indicates severe mode collapse (deterministic generation), while an APD closer to the Ground Truth (GT) indicates healthy mode coverage.

2. Topological Consistency via Link Prediction (AUC): To directly measure how well the generated attributes reconstruct the topology, we freeze the completed attributes $\hat{\mathbf{X}}$ and train a simple inner-product decoder to predict the existing graph edges. A higher AUC means the attributes inherently preserve stronger structural information.

*Table 5.* Comparison of diversity & topological consistency.

| Method | Diversity: APD (↑) | Topology: AUC (↑) |
|---|---|---|
| Ground Truth | 1.18 | 92.5% |
| SAT | 0.18 | 79.2% |
| AMER | 0.20 | 78.6% |
| WGNN | 0.24 | 81.5% |
| **GLAD** | 1.12 | 88.4% |

As shown in the Table 5, the adversarial baselines (SAT, AMER) exhibit an APD near 0.2. This directly proves our hypothesis: under ambiguous missing-feature settings, GAN-based models collapse to generating nearly identical, deterministic completions regardless of the input noise. In contrast, GLAD achieves an APD of 1.12, which closely aligns with the inherent variance of the true data distribution (1.18). This directly validates that our diffusion-based approach successfully captures multiple plausible completions. GLAD achieves an 88.4% Link Prediction AUC, outperforming WGNN by a large margin (+6.9%). This provides direct quantitative proof that our backward alignment mechanism explicitly enforces structural consistency beyond what adversarial alignment can achieve.

## 5. Conclusion

In this paper, we presented GLAD, a novel generative framework for node attribute completion on graphs via latent diffusion models. By shifting the paradigm from adversarial matching to structure-conditioned diffusion, GLAD effectively overcomes the inherent training instabilities and mode collapse issues prevalent in GAN-based predecessors like SAT. Our key technical contribution—the bidirectional alignment mechanism—bridges the gap between topological reconstruction and attribute generation, ensuring that the recovered features are not only diverse but also strictly consistent with the underlying graph structure. Theoretically, we demonstrated that GLAD optimizes a tighter variational lower bound on the joint distribution, providing a principled guarantee for superior mode coverage. Empirically, GLAD achieved state-of-the-art performance across diverse benchmarks, particularly in complex real-world traffic sensor networks where capturing multi-modal uncertainty is crucial. We believe that GLAD sets a new milestone for generative graph imputation and provides a robust foundation for building reliable GNN-based systems in the presence of incomplete data. Future work will explore the extension of GLAD to heterogeneous graph structures and its scalability to web-scale social networks.

## Acknowledgements

This work was partly supported by the National Key Research and Development Program of China under Grant 2023YFB3002201, the National Natural Science Foundation of China under Grant 72342026, and Fundamental Research Funds for the Central Universities under Grant 2024-6-ZD-02. Dr. Yang Zhang of University of North Texas received no financial support for this work from the above grants or any other external projects. His contribution was made independently as part of his academic research. The authors also sincerely acknowledge the valuable collaboration and insightful discussions contributed by colleagues from the participating universities.

## Impact Statement

This work introduces a generative framework for recovering missing data in graph-structured systems, which has significant implications for both the efficiency and reliability of AI-driven infrastructure. The most direct application of our work lies in Urban Intelligence and Sustainability. By accurately imputing missing traffic sensor data, GLAD enables more precise Estimated Time of Arrival (ETA) predictions and better congestion management. This leads to reduced carbon emissions from idling vehicles and improves the efficiency of public transit systems.

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

## A. Dataset Description and Statistics

Table 6 summarizes the key statistics of four widely-used graph datasets.

1) **Cora** (McCallum et al., 2000) is a widely-used *citation network* benchmark for semi-supervised node classification. Nodes correspond to scientific publications and edges denote citation links between papers. Node attributes are sparse bag-of-words features extracted from paper text, and node labels indicate the research topic/category. In our setting, Cora contains 2,708 nodes and 5,278 edges, with a graph density 0.07%; the average and maximum in-degree are 3.90 and 168, respectively. The feature dimension is 1,433, with an average of 18.17 non-zero entries per node (i.e., feature sparsity), and there are 7 classes.

2) **CiteSeer** (Sen et al., 2008) is another standard *citation network* dataset with the same semantics as Cora: nodes are documents, edges are citations, and node attributes are bag-of-words text features with topic labels. It includes 3,327 nodes and 4,228 edges, with density 0.04%; the average and maximum in-degree are 2.77 and 99. CiteSeer has 3,703-dimensional sparse features with 31.6 average non-zeros per node, and 6 classes.

3) **Amazon-Computer** (Shchur et al., 2018) is a *co-purchase network* where nodes represent products and edges indicate that two products are frequently bought together. Node features are derived from product reviews (bag-of-words), and labels correspond to product categories. The graph contains 13,752 nodes and 245,861 edges (density 0.13%), with average/max in-degree 35.76/2,992. It has 767-dimensional sparse features with 267.23 average non-zeros per node, and 10 classes.

4) **Amazon-Photo** (Shchur et al., 2018) follows the same *co-purchase* construction as Amazon-Computer (products as nodes; "also bought together" relations as edges; review-based bag-of-words node features; category labels). It includes 7,650 nodes and 119,081 edges (density 0.20%), with average/max in-degree 31.13/1,434. The feature dimension is 745, with 258.81 average non-zeros per node, and 8 classes.

*Table 6.* Basic dataset statistics of Cora, Citeseer, Amazon-Computer, and Amazon-Photo (# denotes the number of).

|  | Cora | Citeseer | Amazon-Computer | Amazon-Photo |
|---|---|---|---|---|
| #Nodes | 2,708 | 3,327 | 13,752 | 7,650 |
| #Edges | 5,278 | 4,228 | 245,861 | 119,081 |
| #Graph_density | 0.07% | 0.04% | 0.13% | 0.20% |
| #Avg_in_degree | 3.90 | 2.77 | 35.76 | 31.13 |
| #Max_in_degree | 168 | 99 | 2,992 | 1,434 |
| #Attribute_dim | 1,433 | 3,703 | 767 | 745 |
| #Avg_hot_num | 18.17 | 31.6 | 267.23 | 258.81 |
| #Class | 7 | 6 | 10 | 8 |

To test robustness in industrial scenarios, we utilize three urban traffic datasets (Neun et al., 2023a;b) collected from **London (LND)**, **Madrid (MAD)**, and **Melbourne (MEL)**. Each city is represented as a directed road graph, where nodes correspond to road junctions (and associated detector locations), and edges correspond to directed road segments. The raw traffic signals are collected by *stationary vehicle counters* deployed by local transportation authorities, resulting in spatially sparse observations over the full road network.

**Data modality and supervision.** The observed signals are time-stamped traffic counts from stationary detectors. Depending on the task setting, the supervision is derived from aggregated travel-time/ETA measurements over sampled *supersegments* (paths consisting of multiple road segments). We report dataset-level summary statistics including the number of days, counters, graph size (nodes/edges), the number of supersegments, and the min/max/mean ETA values (in seconds).

**City-specific statistics.** Table 7 summarizes the basic statistics of the three datasets. Overall, the three cities exhibit comparable time coverage but different network scales and traffic dynamics, providing a diverse testbed for evaluating the robustness and generalization of our method across heterogeneous urban environments.

*Table 7.* Basic statistics of the three urban traffic datasets (London, Madrid, Melbourne).

|  | London (LND) | Madrid (MAD) | Melbourne (MEL) |
| --- | --- | --- | --- |
| Period | Jul. 1st 2019∼Jan. 31st 2020 | Jun. 1st 2021∼Dec. 31st 2021 | Jun. 1st 2020∼Dec. 30th 2020 |
| #Days | 110 | 109 | 106 |
| #Detectors | 3,751 | 3,875 | 3,982 |
| #Nodes $|\mathcal{N}|$ | 59,110 | 63,397 | 49,510 |
| #Edges $|\mathcal{E}|$ | 132,414 | 121,902 | 94,871 |
| #Supersegments $|\mathcal{S}|$ | 4,012 | 3,969 | 3,246 |
| Min(s) of ETA | 16.2 | 15.7 | 19.1 |
| Max(s) of ETA | 3600.0 | 3600.0 | 3600.0 |
| Mean(s) of ETA | 393.3 | 222.2 | 343.8 |

## B. Exploratory Data Analysis and Visualization

In this section, we provide a visualization analysis of the datasets used in our experiments, covering both standard citation/co-purchase networks and real-world traffic sensor networks.

### B.1. Standard Benchmarks: Topological and Label Characteristics

The class imbalance observed in Figure 5 justifies the robustness of GLAD across varying label densities. All four datasets exhibit a pronounced **long-tail distribution** for in-degree (also known as a power-law distribution). Traditional propagation-based methods (e.g., GINN, GCN) often fail on these long-tailed graphs because they rely on dense local neighborhoods. In contrast, GLAD's latent diffusion process, enhanced by bidirectional alignment, effectively models the conditional distribution of attributes even for sparsely connected nodes. By maximizing the mutual information between attributes and the global structure, GLAD leverages the influence of hub nodes to guide the completion of "tail" nodes, ensuring robust performance across the entire degree spectrum.

As shown in Figure 6, we also visualize the sub-graph structures for Cora, Citeseer, Amazon-Computer, and Amazon-Photo.

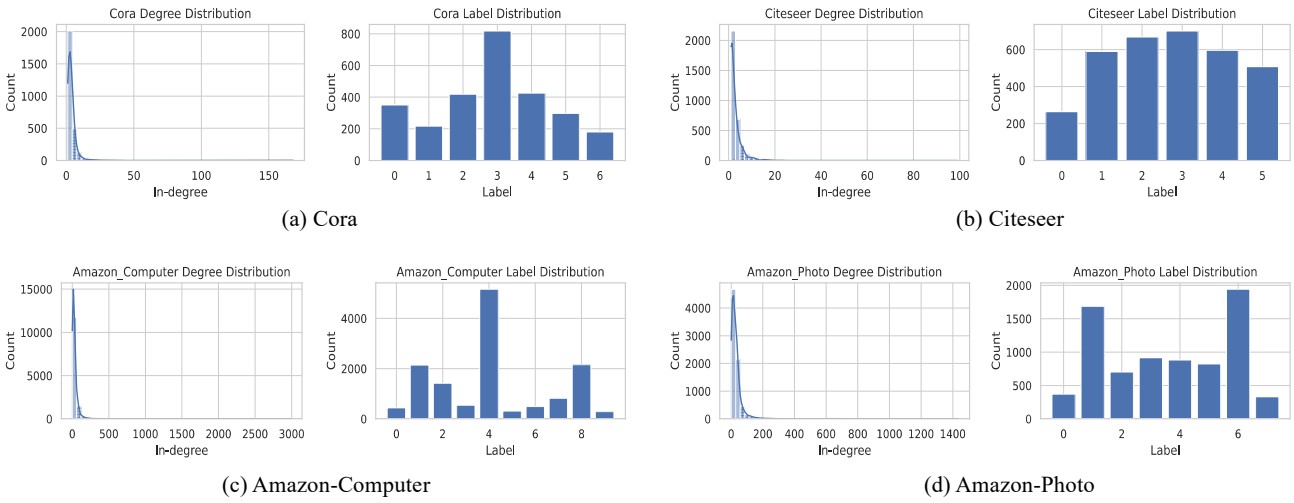

*Figure 5.* In-degree distributions and label characteristics of standard benchmarks.

### B.2. Real-World Traffic Networks: Spatial and Flow Visualization

For the urban traffic datasets (London, Madrid, Melbourne), we visualize the road network topology in Figure 7 and the temporal traffic flow patterns in Figure 8.

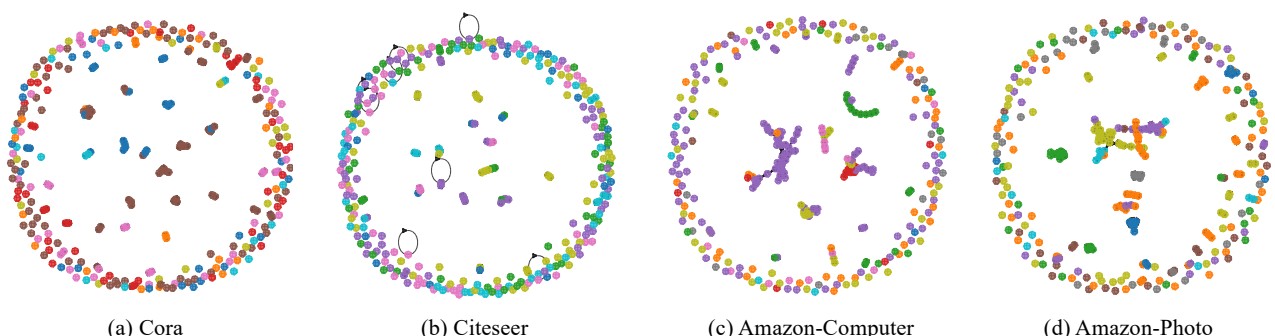

| (a) Cora | (b) Citeseer | (c) Amazon-Computer | (d) Amazon-Photo |

*Figure 6.* Subgraph structure of standard benchmarks (Cora, Citeseer, Amazon-Computer, and Amazon-Photo).

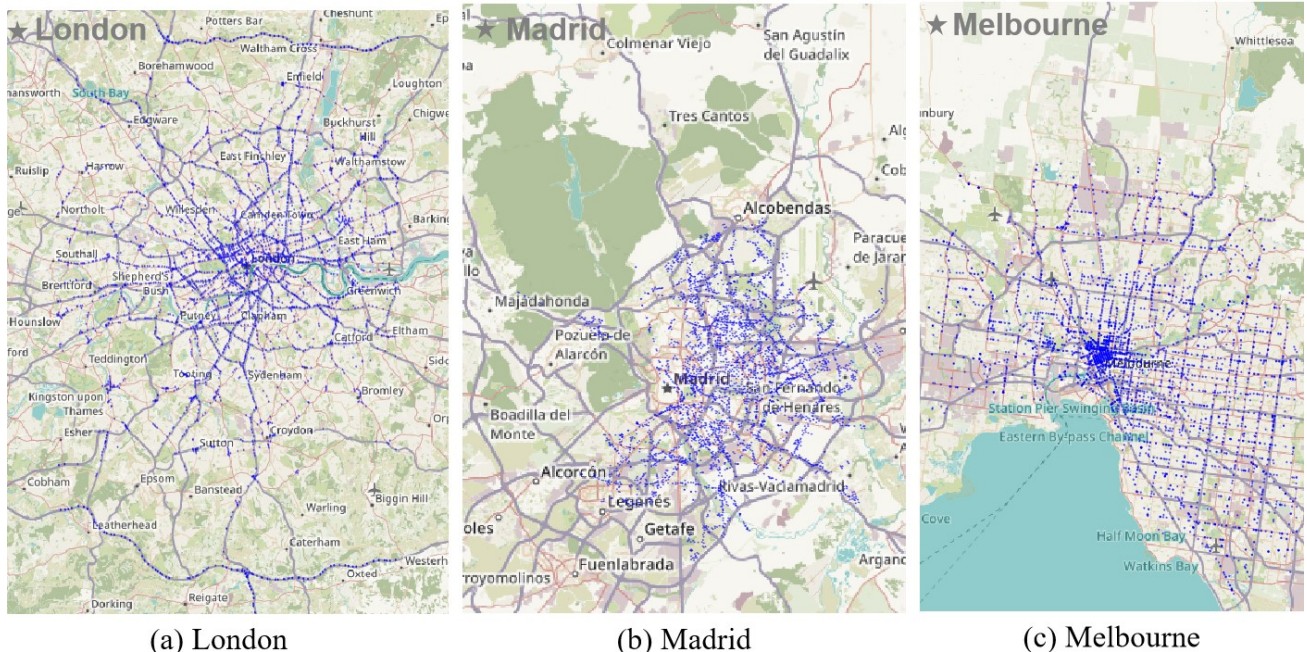

| (a) London | (b) Madrid | (c) Melbourne |

*Figure 7.* Visualization of the complex connectivity of sensor nodes in metropolitan areas.

## C. Evaluation Metrics

To provide a comprehensive evaluation of GLAD, we employ various metrics tailored to different tasks, ranging from attribute reconstruction to downstream predictive tasks.

### C.1. Attribute Recovery Metrics

For the direct quality assessment of imputed attributes, we treat the task as a ranking problem for sparse datasets (e.g., Cora, Citeseer) and a reconstruction problem for dense datasets.

- **Recall@k**: This metric measures the proportion of the top-$k$ ground truth features that are successfully recovered in the model's top-$k$ predicted features for each node:

$$\text{Recall@k} = \frac{1}{|V_{miss}|} \sum_{i \in V_{miss}} \frac{|\text{TopK}(\mathbf{x}_i) \cap \text{TopK}(\hat{\mathbf{x}}_i)|}{k}, \tag{7}$$

  where $\text{TopK}(\cdot)$ denotes the set of indices of the $k$ largest values in the attribute vector.

- **NDCG@k**: Normalized Discounted Cumulative Gain (NDCG) accounts for the ranking position of the recovered

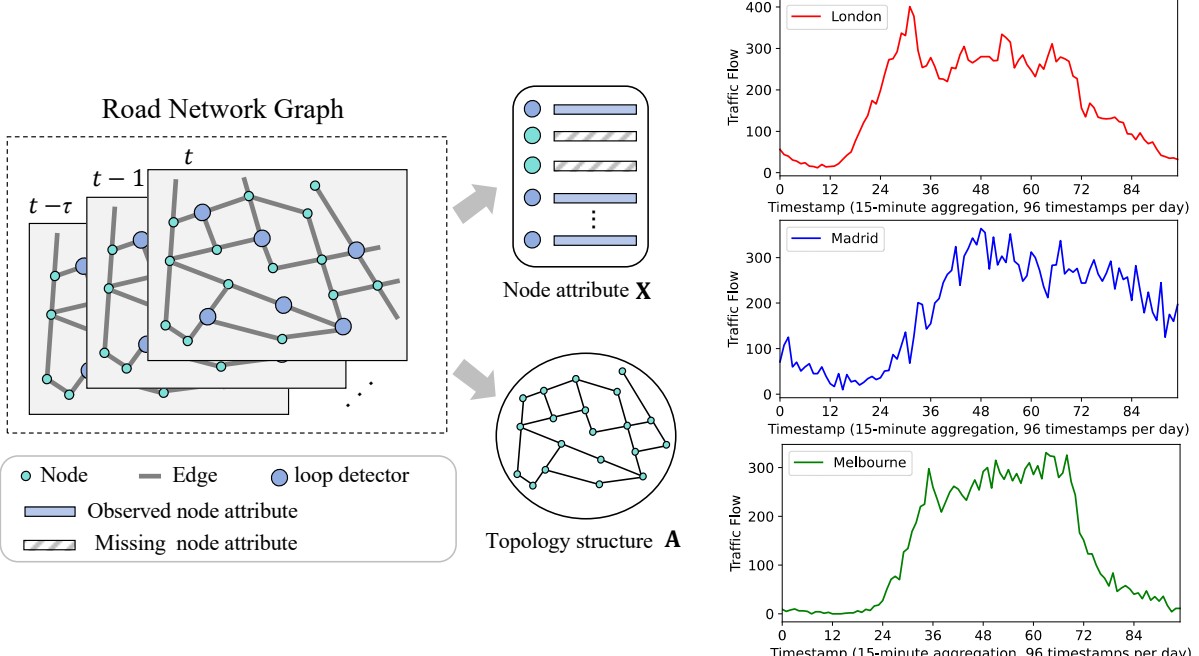

(a) Road network with missing attributes     (b) Observed flow distributions with non-linear feature

*Figure 8.* The traffic flow distributions of the high-dimensional and non-linear attributes.

features, assigning higher weights to features correctly placed at the top of the list:

$$\text{DCG@k} = \sum_{j=1}^{k} \frac{r_j}{\log_2(j+1)}, \quad \text{NDCG@k} = \frac{\text{DCG@k}}{\text{IDCG@k}}, \tag{8}$$

where $r_j$ is the relevance of the feature at rank $j$, and IDCG is the ideal DCG achieved by perfect ranking.

## C.2. Downstream Task Metrics

For node classification and traffic state classification, we use the standard **Accuracy** metric:

$$\text{Accuracy} = \frac{1}{|V_{test}|} \sum_{i \in V_{test}} \mathbb{I}(y_i = \hat{y}_i), \tag{9}$$

where $\mathbb{I}(\cdot)$ is the indicator function, and $y_i, \hat{y}_i$ are the ground truth and predicted labels, respectively.

## C.3. Numerical Regression Metrics (ETA Task)

For the Estimated Time of Arrival (ETA) task, where attributes are continuous numerical values (traffic speeds), we employ three standard regression metrics:

- **Mean Absolute Error (MAE)**: Measures the average magnitude of errors in the predictions:

$$\text{MAE} = \frac{1}{N} \sum_{i=1}^{N} |x_i - \hat{x}_i|, \tag{10}$$

- **Root Mean Square Error (RMSE)**: Provides a quadratic scoring rule that is particularly sensitive to large errors or outliers:

$$\text{RMSE} = \sqrt{\frac{1}{N} \sum_{i=1}^{N} (x_i - \hat{x}_i)^2}, \tag{11}$$

- **Mean Absolute Percentage Error (MAPE)**: Measures the relative error as a percentage, which is useful for comparing performance across datasets with different scales:

$$\text{MAPE} = \frac{100\%}{N} \sum_{i=1}^{N} \left| \frac{x_i - \hat{x}_i}{x_i} \right|, \tag{12}$$

Consistent with standard practices in traffic forecasting, lower values for MAE, RMSE, and MAPE, and higher values for Recall@k, NDCG@k, and Accuracy indicate superior performance.

## D. Detailed Training and Inference Algorithms

Algorithm 1 details the training procedure with bidirectional alignment. It integrates three synergistic phases: (i) mapping graph topology and attributes into a decoupled latent space; (ii) performing latent diffusion with conditional denoising, where a condition-dropout mechanism is employed to support Classifier-Free Guidance (CFG); and (iii) enforcing bidirectional consistency by reconstructing the graph structure from the estimated latent attributes. The total objective balances generative fidelity and topological alignment, ensuring that the model captures the intricate joint distribution of graph data.

---

**Algorithm 1** GLAD Training with Bidirectional Alignment

---

**Input:** Attribute-missing graph $\mathcal{G} = (\mathbf{A}, \mathbf{X})$, structure encoder $E_{struct}$, attribute encoder $E_{attr}$, denoising network $\epsilon_\theta$, structure decoder $D_{struct}$, hyperparameter $\lambda$, guidance dropout $p_{uncond}$.
**Initialize:** Parameters $\theta$, encoders/decoders, and variance schedule $\{\beta_t\}_{t=1}^{T}$.
**repeat**
    *# Phase 1: Decoupled Latent Encoding*
    $\mathbf{Z}_{struct} \leftarrow E_{struct}(\mathbf{A})$ {Topological embedding}
    $\mathbf{Z}_{attr}^{(0)} \leftarrow E_{attr}(\mathbf{X}_{obs})$ {Attribute latent for observed nodes}
    *# Phase 2: Latent Diffusion Training*
    Sample $t \sim \text{Uniform}(\{1, \ldots, T\})$ and $\epsilon \sim \mathcal{N}(\mathbf{0}, \mathbf{I})$.
    $\mathbf{z}_t \leftarrow \sqrt{\bar{\alpha}_t}\mathbf{Z}_{attr}^{(0)} + \sqrt{1 - \bar{\alpha}_t}\epsilon$ {Forward diffusion}
    *# Classifier-Free Guidance Dropout*
    $\mathbf{c} \leftarrow \mathbf{Z}_{struct}$ with probability $1 - p_{uncond}$, else $\mathbf{c} \leftarrow \emptyset$.
    $\mathcal{L}_{diff} \leftarrow \|\epsilon - \epsilon_\theta(\mathbf{z}_t, t, \mathbf{c})\|^2$ {Diffusion loss on observed nodes}
    *# Phase 3: Backward Topological Alignment*
    $\hat{\mathbf{z}}_0 \leftarrow \frac{1}{\sqrt{\bar{\alpha}_t}}(\mathbf{z}_t - \sqrt{1 - \bar{\alpha}_t}\epsilon_\theta(\mathbf{z}_t, t, \mathbf{c}))$ {One-step reconstruction of $\mathbf{z}_0$}
    $\hat{\mathbf{A}} \leftarrow D_{struct}(\hat{\mathbf{z}}_0)$ {Structure reconstruction from attribute latents}
    $\mathcal{L}_{align} \leftarrow \|\mathbf{A} - \hat{\mathbf{A}}\|^2$ {Backward alignment loss}
    *# Total Objective Optimization*
    $\mathcal{L}_{total} \leftarrow \mathcal{L}_{diff} + \lambda\mathcal{L}_{align}$
    Update $\theta$ via $\nabla_\theta \mathcal{L}_{total}$.
**until** convergence

---

Algorithm 2 describes the structure-aware conditional sampling process. Starting from standard Gaussian noise, the algorithm performs iterative stochastic denoising guided by the learned structural latent. The key innovation lies in the Structure-Aware Classifier-Free Guidance (CFG): at each timestep, the model calculates both conditional and unconditional noise estimates, extrapolating them via a guidance scale $w$ to steer the generative process toward structurally relevant regions. This ensures the completed attributes are not only plausible but also strictly grounded in the underlying graph topology.

Algorithm 3 elaborates on the core computational unit of the denoising network ($\epsilon_\theta$) in GLAD. The block first integrates temporal information via sinusoidal embeddings, followed by a self-attention layer to capture intra-attribute dependencies. Crucially, it employs a Multi-Head Cross-Attention (MHCA) mechanism, where structural latents serve as Keys and Values to guide the Queries from the attribute latent space. This architecture enables the model to dynamically steer the attribute generation based on the topological context at each denoising step, providing a fundamental mechanism for the rigorous alignment of structure and attributes.

---

**Algorithm 2** GLAD Structure-Aware Conditional Sampling

---

**Input:** Adjacency matrix $\mathbf{A}$, guidance scale $w$, trained GLAD components.
$\mathbf{Z}_{struct} \leftarrow E_{struct}(\mathbf{A})$.
Sample $\mathbf{z}_T \sim \mathcal{N}(\mathbf{0}, \mathbf{I})$ for missing nodes $V_{miss}$.
**for** $t = T$ **to** 1 **do**
    *# Dual Noise Estimation for CFG*
    $\epsilon_\theta^{cond} \leftarrow \epsilon_\theta(\mathbf{z}_t, t, \mathbf{Z}_{struct})$ {Structure-conditioned estimate}
    $\epsilon_\theta^{uncond} \leftarrow \epsilon_\theta(\mathbf{z}_t, t, \emptyset)$ {Unconditional estimate}
    $\tilde{\epsilon}_\theta \leftarrow (1+w)\epsilon_\theta^{cond} - w\epsilon_\theta^{uncond}$ {Guided noise extrapolation}
    *# Reverse Diffusion Step*
    Sample $\zeta \sim \mathcal{N}(\mathbf{0}, \mathbf{I})$ if $t > 1$, else $\zeta = \mathbf{0}$.
    $\mathbf{z}_{t-1} \leftarrow \frac{1}{\sqrt{\alpha_t}}\left(\mathbf{z}_t - \frac{1-\alpha_t}{\sqrt{1-\bar{\alpha}_t}}\tilde{\epsilon}_\theta\right) + \sigma_t\zeta$
**end for**
$\hat{\mathbf{X}}_{miss} \leftarrow D_{attr}(\mathbf{z}_0)$.
**Return:** Completed attribute matrix $\hat{\mathbf{X}} = [\mathbf{X}_{obs}; \hat{\mathbf{X}}_{miss}]$.

---

Algorithm 4 details the scalable alignment strategy of GLAD for large-scale sparse graphs. To circumvent the $O(N^2)$ complexity of full adjacency reconstruction, we employ a negative sampling scheme that computes the alignment loss over a subset of sampled edges and non-edges.

---

**Algorithm 3** Structure-Conditioned Latent Denoising

---

**Input:** Noisy attribute latents $\mathbf{z}_t$, structure latents $\mathbf{Z}_{struct}$, timestep $t$, number of heads $H$.
**Output:** Predicted noise $\epsilon_\theta$.
*# 1. Temporal Embedding*
$\tau \leftarrow \text{MLP}(\text{SinusoidalPositionalEmbedding}(t))$
*# 2. Self-Attention (Attribute Introspection)*
$\mathbf{z}'_t \leftarrow \text{LayerNorm}(\mathbf{z}_t + \tau)$
$\mathbf{Q}_{sa}, \mathbf{K}_{sa}, \mathbf{V}_{sa} \leftarrow \text{Linear}_{q,k,v}(\mathbf{z}'_t)$
$\mathbf{z}_{sa} \leftarrow \text{MultiHeadAttention}(\mathbf{Q}_{sa}, \mathbf{K}_{sa}, \mathbf{V}_{sa})$
*# 3. Structure-Aware Cross-Attention (Structural Guidance)*
$\mathbf{Q}_{ca} \leftarrow \text{Linear}_q(\mathbf{Z}_{sa})$ {Queries from attribute space}
$\mathbf{K}_{ca} \leftarrow \text{Linear}_k(\mathbf{Z}_{struct})$ {Keys from structural space}
$\mathbf{V}_{ca} \leftarrow \text{Linear}_v(\mathbf{Z}_{struct})$ {Values from structural space}
**for** $h = 1$ **to** $H$ **do**
    $\text{Attn}_h \leftarrow \text{Softmax}\left(\frac{\mathbf{Q}_{ca}^{(h)}(\mathbf{K}_{ca}^{(h)})^\top}{\sqrt{d/H}}\right)$
    $\text{Head}_h \leftarrow \text{Attn}_h \mathbf{V}_{ca}^{(h)}$
**end for**
$\mathbf{z}_{ca} \leftarrow \text{Concat}(\text{Head}_1, \dots, \text{Head}_H)\mathbf{W}_O$
*# 4. Feed-Forward and Output*
$\epsilon_\theta \leftarrow \text{LayerNorm}(\mathbf{z}_{sa} + \mathbf{z}_{ca})$
$\epsilon_\theta \leftarrow \text{MLP}(\epsilon_\theta)$
**Return:** $\epsilon_\theta$

---

# E. Proof of Proposition and Theorem

In this section, we provide detailed derivations and proofs for the theoretical claims made in Section 3.6 of the main paper. We first define the necessary notations. Let $\mathbf{A}$ denote the graph adjacency matrix and $\mathbf{X}$ denote the node attributes. In GLAD, we operate in a decoupled latent space, where $\mathbf{Z}_{struct}$ and $\mathbf{Z}_{attr}$ are the latent representations for structure and attributes, respectively. The true data distribution is denoted as $p_{data}(\mathbf{Z}_{attr}|\mathbf{Z}_{struct})$, and the model distribution is $p_\theta(\mathbf{Z}_{attr}|\mathbf{Z}_{struct})$.

---

**Algorithm 4** Scalable Topological Alignment via Negative Sampling

---

**Input:** Adjacency $\mathbf{A}$, attribute latents $\hat{\mathbf{z}}_0$, sample size $M$.
$\mathcal{E}^+ \leftarrow$ Randomly sample $M$ positive edges from $\{(i,j) \mid A_{ij} = 1\}$.
$\mathcal{E}^- \leftarrow$ Randomly sample $M$ negative pairs from $\{(i,j) \mid A_{ij} = 0\}$.
$\mathcal{L}_{align} \leftarrow \sum_{(i,j) \in \mathcal{E}^+ \cup \mathcal{E}^-} \text{BinaryCrossEntropy}(D_{struct}(\hat{\mathbf{z}}_{0,i}, \hat{\mathbf{z}}_{0,j}), A_{ij})$.
**Return:** Scaled alignment loss $\mathcal{L}_{align}$.

---

### E.1. Proof of Proposition 3.1

**Proposition 3.1** (Training Objective Comparison). *Adversarial alignment methods implicitly minimize the Jensen-Shannon (JS) divergence between the latent distributions of structure and attributes. In contrast, optimizing the diffusion loss $\mathcal{L}_{diff}$ in GLAD is equivalent to maximizing the Evidence Lower Bound (ELBO) of the data log-likelihood, which effectively minimizes an upper bound of the forward Kullback-Leibler (KL) divergence, $D_{KL}(p_{data}(\mathbf{X}|\mathbf{A})\|p_\theta(\mathbf{X}|\mathbf{A}))$.*

*Proof.* **Part 1: Adversarial Alignment and JS Divergence.** Recall that standard GAN-based methods (e.g., SAT) employ a discriminator $D$ to distinguish between samples from the true distribution $p_{data}(x)$ and the generator distribution $p_\theta(x)$. The minimax objective is:

$$\min_G \max_D V(D, G) = \mathbb{E}_{x \sim p_{data}}[\log D(x)] + \mathbb{E}_{x \sim p_\theta}[\log(1 - D(x))]. \tag{13}$$

As shown by (Goodfellow et al., 2014), for the optimal discriminator $D_G^*(x) = \frac{p_{data}(x)}{p_{data}(x) + p_\theta(x)}$, the generator's objective reduces to minimizing the Jensen-Shannon divergence:

$$C(G) = -\log 4 + 2 \cdot D_{JS}(p_{data}\|p_\theta), \tag{14}$$

where $D_{JS}(p\|q) = \frac{1}{2}D_{KL}(p\|\frac{p+q}{2}) + \frac{1}{2}D_{KL}(q\|\frac{p+q}{2})$. Thus, adversarial training seeks to align distributions by minimizing the JS divergence (Menéndez et al., 1997; Briët & Harremoës, 2009).

**Part 2: Diffusion Models and ELBO/Forward KL.** GLAD employs a Latent Diffusion Model. The training objective is to maximize the log-likelihood of the data $p_\theta(\mathbf{Z}_{attr}|\mathbf{Z}_{struct})$. Since the exact likelihood is intractable, we optimize the Variational Lower Bound (ELBO). Let $\mathbf{z}_0 = \mathbf{Z}_{attr}$ and $\mathbf{c} = \mathbf{Z}_{struct}$. The diffusion process introduces latent variables $\mathbf{z}_{1:T}$. The ELBO is derived as:

$$
\begin{aligned}
\log p_\theta(\mathbf{z}_0|\mathbf{c}) &= \log \int p_\theta(\mathbf{z}_{0:T}|\mathbf{c}) d\mathbf{z}_{1:T} \\
&= \log \mathbb{E}_{q(\mathbf{z}_{1:T}|\mathbf{z}_0)} \left[ \frac{p_\theta(\mathbf{z}_{0:T}|\mathbf{c})}{q(\mathbf{z}_{1:T}|\mathbf{z}_0)} \right] \\
&\geq \mathbb{E}_{q(\mathbf{z}_{1:T}|\mathbf{z}_0)} \left[ \log \frac{p_\theta(\mathbf{z}_{0:T}|\mathbf{c})}{q(\mathbf{z}_{1:T}|\mathbf{z}_0)} \right] \quad \text{(Jensen's Inequality)} \\
&= \mathbb{E}_q \left[ \log p_\theta(\mathbf{z}_T|\mathbf{c}) + \sum_{t=1}^T \log \frac{p_\theta(\mathbf{z}_{t-1}|\mathbf{z}_t, \mathbf{c})}{q(\mathbf{z}_t|\mathbf{z}_{t-1})} \right].
\end{aligned}
\tag{15}
$$

Typically, diffusion training objectives (like the simple MSE loss $\mathcal{L}_{simple} = \|\epsilon - \epsilon_\theta\|^2$) are reweighted versions of this ELBO (Ho et al., 2020). Crucially, maximizing the log-likelihood (or ELBO) is equivalent to minimizing the Forward KL divergence between the data distribution and the model distribution:

$$
\begin{aligned}
\min_\theta D_{KL}(p_{data}(\mathbf{z}_0|\mathbf{c})\|p_\theta(\mathbf{z}_0|\mathbf{c})) &= \min_\theta \mathbb{E}_{\mathbf{z}_0 \sim p_{data}} \left[ \log \frac{p_{data}(\mathbf{z}_0|\mathbf{c})}{p_\theta(\mathbf{z}_0|\mathbf{c})} \right] \\
&= \min_\theta \left( -\mathbb{E}_{p_{data}}[\log p_\theta(\mathbf{z}_0|\mathbf{c})] + \text{const} \right).
\end{aligned}
\tag{16}
$$

Thus, GLAD optimizes the Forward KL divergence, whereas GANs optimize the JS divergence (which behaves similarly to Reverse KL in terms of mode seeking). □

### E.2. Proof of Theorem 3.2

**Theorem 3.2** (Superior Mode Coverage of GLAD). *Let $\mathcal{M}_{data} = \{X : p_{data}(X|A) > 0\}$ be the support of the true conditional distribution. Assuming the model has sufficient capacity, the optimal distribution $p_\theta^*$ learned by minimizing the forward Kullback-Leibler (KL) divergence satisfies $\mathcal{M}_{data} \subseteq \mathrm{supp}(p_\theta^*)$ almost everywhere, i.e., $p_\theta^*(X|A) > 0$ for almost all $X \in \mathcal{M}_{data}$. Conversely, optimizing the Jensen-Shannon (JS) divergence imposes only a bounded penalty for missing modes, theoretically permitting mode collapse where $\mathrm{supp}(p_\theta) \subset \mathcal{M}_{data}$.*

*Proof.* For simplicity of notation, let $x$ denote the attribute $X$, and we omit the conditioning on $A$ as the proof holds for any fixed structure $A$. Let $\mathcal{M}_{data} = \{x : p_{data}(x) > 0\}$.

**Part 1: Forward KL Divergence guarantees Support Coverage.** The objective optimized by GLAD is an upper bound on the Forward KL divergence:

$$D_{KL}(p_{data} \parallel p_\theta) = \int_{\mathcal{M}_{data}} p_{data}(x) \log \frac{p_{data}(x)}{p_\theta(x)} dx \tag{17}$$

Assume, for the sake of contradiction, that the optimal learned model $p_\theta^*$ fails to cover a subset of the true data support. That is, there exists a measurable set $\mathcal{S} \subseteq \mathcal{M}_{data}$ with strictly positive probability mass under the data distribution, $P_{data}(\mathcal{S}) = \delta > 0$, such that the model assigns zero (or infinitesimally small) probability to $\mathcal{S}$: $\lim_{\epsilon \to 0^+} \int_{\mathcal{S}} p_\theta(x) dx = \epsilon$.

The KL divergence can be decomposed as:

$$D_{KL}(p_{data} \parallel p_\theta) = \int_{\mathcal{M}_{data} \setminus \mathcal{S}} p_{data}(x) \log \frac{p_{data}(x)}{p_\theta(x)} dx + \int_{\mathcal{S}} p_{data}(x) \log \frac{p_{data}(x)}{p_\theta(x)} dx \tag{18}$$

By the Log-Sum Inequality, the integral over the dropped mode $\mathcal{S}$ is lower-bounded:

$$\int_{\mathcal{S}} p_{data}(x) \log \frac{p_{data}(x)}{p_\theta(x)} dx \geq P_{data}(\mathcal{S}) \log \frac{P_{data}(\mathcal{S})}{P_\theta(\mathcal{S})} = \delta \log \frac{\delta}{\epsilon} \tag{19}$$

As $\epsilon \to 0$ (i.e., $p_\theta(x) \to 0$ for $x \in \mathcal{S}$), we have $\delta \log(\delta/\epsilon) \to +\infty$. Therefore, if $p_\theta^*$ fails to cover any set of non-zero measure in $\mathcal{M}_{data}$, the Forward KL divergence diverges to infinity. To achieve a finite minimum in optimization, the optimal model *must* satisfy $p_\theta^*(x) > 0$ almost everywhere on $\mathcal{M}_{data}$. Thus, $\mathcal{M}_{data} \subseteq \mathrm{supp}(p_\theta^*)$, explicitly guaranteeing mode coverage.

**Part 2: JS Divergence allows Mode Collapse.** In adversarial methods (e.g., SAT), the optimal discriminator leads the generator to minimize the JS divergence. Let $M = \frac{1}{2}(p_{data} + p_\theta)$.

$$D_{JS}(p_{data} \parallel p_\theta) = \frac{1}{2} D_{KL}(p_{data} \parallel M) + \frac{1}{2} D_{KL}(p_\theta \parallel M) \tag{20}$$

Now, consider the same scenario where the model $p_\theta$ completely drops the mode $\mathcal{S} \subseteq \mathcal{M}_{data}$ (i.e., $p_\theta(x) = 0, \forall x \in \mathcal{S}$). On the set $\mathcal{S}$, the mixture distribution is $M(x) = \frac{1}{2} p_{data}(x)$. The penalty contributed by this dropped mode to the JS divergence is:

$$\frac{1}{2} \int_{\mathcal{S}} p_{data}(x) \log \frac{p_{data}(x)}{p_{data}(x)/2} dx + \frac{1}{2} \int_{\mathcal{S}} 0 \cdot \log \frac{0}{M(x)} dx = \frac{1}{2} \int_{\mathcal{S}} p_{data}(x) \log 2 \, dx = \frac{\delta \log 2}{2} \tag{21}$$

Unlike the Forward KL divergence which penalizes the missing mode with $+\infty$, the JS divergence imposes a strictly bounded penalty ($\leq \frac{1}{2} \log 2$) for completely missing a data mode. Because this penalty is finite and relatively small, the optimization landscape permits the model to ignore difficult modes in $\mathcal{M}_{data}$ (mode collapse) if doing so lowers the overall loss elsewhere, meaning it is not guaranteed that $\mathcal{M}_{data} \subseteq \mathrm{supp}(p_\theta)$. $\square$

### E.3. Proof of Theorem 3.3

**Theorem 3.3** (Information Consistency via Mutual Information). *Let $\mathcal{L}_{align} = \|\mathbf{A} - \hat{\mathbf{A}}\|^2$ be the backward topological alignment loss, where $\hat{\mathbf{A}} = D_{struct}(\mathbf{Z}_{attr})$. Minimizing $\mathcal{L}_{align}$ is equivalent to maximizing a variational lower bound on the mutual information $I(\mathbf{A}; \mathbf{Z}_{attr})$ between the graph topology $\mathbf{A}$ and the generated attribute latents $\mathbf{Z}_{attr}$.*

*Proof.* We start with the definition of Mutual Information between the adjacency matrix $\mathbf{A}$ and the attribute latents $\mathbf{Z}_{attr}$:

$$I(\mathbf{A}; \mathbf{Z}_{attr}) = H(\mathbf{A}) - H(\mathbf{A}|\mathbf{Z}_{attr}), \tag{22}$$

where $H(\cdot)$ denotes the entropy. Since the ground truth graph structure $\mathbf{A}$ is fixed during training, $H(\mathbf{A})$ is a constant. Therefore, maximizing mutual information is equivalent to minimizing the conditional entropy $H(\mathbf{A}|\mathbf{Z}_{attr})$:

$$\max I(\mathbf{A}; \mathbf{Z}_{attr}) \iff \min H(\mathbf{A}|\mathbf{Z}_{attr}) = \min \mathbb{E}_{p(\mathbf{A}, \mathbf{Z}_{attr})}[- \log p(\mathbf{A}|\mathbf{Z}_{attr})]. \tag{23}$$

However, the true posterior $p(\mathbf{A}|\mathbf{Z}_{attr})$ is unknown. We introduce a variational approximation $q_\phi(\mathbf{A}|\mathbf{Z}_{attr})$, parameterized by our structure reconstruction decoder $D_{struct}$. By the non-negativity of KL divergence $D_{KL}(p(\mathbf{A}|\mathbf{Z}_{attr})\|q_\phi(\mathbf{A}|\mathbf{Z}_{attr})) \geq 0$, we have:

$$\begin{aligned} \mathbb{E}_{p(\mathbf{A}, \mathbf{Z}_{attr})}[- \log p(\mathbf{A}|\mathbf{Z}_{attr})] &\leq \mathbb{E}_{p(\mathbf{A}, \mathbf{Z}_{attr})}[- \log q_\phi(\mathbf{A}|\mathbf{Z}_{attr})] \\ &= \mathbb{E}_{\mathbf{Z}_{attr} \sim p(\mathbf{Z}_{attr})}[\mathcal{L}_{recon}(\mathbf{A}, \mathbf{Z}_{attr})]. \end{aligned} \tag{24}$$

Now, we define the form of the variational distribution $q_\phi(\mathbf{A}|\mathbf{Z}_{attr})$. If we assume a Gaussian observation model for the continuous relaxation of the adjacency matrix (or the logits):

$$q_\phi(\mathbf{A}|\mathbf{Z}_{attr}) = \mathcal{N}(\mathbf{A}; D_{struct}(\mathbf{Z}_{attr}), \sigma^2 \mathbf{I}), \tag{25}$$

then the negative log-likelihood becomes:

$$- \log q_\phi(\mathbf{A}|\mathbf{Z}_{attr}) \propto \frac{1}{2\sigma^2} \|\mathbf{A} - D_{struct}(\mathbf{Z}_{attr})\|^2 + \text{const.} \tag{26}$$

Ignoring the constants, minimizing the negative log-likelihood is equivalent to minimizing the Mean Squared Error (MSE) reconstruction loss:

$$\mathcal{L}_{align} = \|\mathbf{A} - D_{struct}(\mathbf{Z}_{attr})\|^2. \tag{27}$$

(Note: If $\mathbf{A}$ is treated as binary, a Bernoulli assumption (Lindley & Phillips, 1976; Korobkov, 2011; Ristic et al., 2013) leads to Binary Cross Entropy loss, which follows the same logic).

**Conclusion.** Minimizing the backward alignment loss $\mathcal{L}_{align}$ minimizes the upper bound of the conditional entropy $H(\mathbf{A}|\mathbf{Z}_{attr})$, which in turn maximizes the variational lower bound of the mutual information $I(\mathbf{A}; \mathbf{Z}_{attr})$. This proves that our alignment mechanism explicitly enforces high information overlap between the generated attributes and the graph topology. $\square$

# F. Additional Experimental Result

## I. The Core Intuition of our GLAD

In our problem, we deal with two modalities: the fixed, discrete graph topology $A \in \{0,1\}^{N \times N}$, and the generated continuous attribute latents $\hat{Z}_{attr} \in \mathbb{R}^{N \times d}$.

Our intuition is to establish an **Information Bottleneck**. To ensure the generated attributes respect the graph structure, we must explicitly maximize their Mutual Information $I(A; \hat{Z}_{attr})$. As proven in our **Theorem 3.3**, minimizing our backward alignment loss precisely maximizes the variational lower bound of this mutual information:

$$\max I(A; \hat{Z}_{attr}) \iff \max \mathbb{E}\left[\log q_\phi(A|\hat{Z}_{attr})\right] \iff \min \mathcal{L}_{align} = \|A - \mathcal{D}_{struct}(\hat{Z}_{attr})\|_F^2$$

By decoding directly to the raw, constant matrix $A$, the topology acts as a **fixed, absolute anchor** for the gradient updates, preventing the generation of topologically impossible features.

## II. Alternatives and Their Mathematical Limitations

With this intuition established, we provide a formal analysis of the alternative designs:

*Alternative A: Latent-to-Latent Matching*

Instead of reconstructing $A$, we could align the generated attribute latents with the structure latents $Z_{struct} = \mathcal{E}_{struct}(A; \phi_S)$, where $\phi_S$ are the trainable parameters of the GNN. The objective would be:

$$\min_{\theta, \phi_S} \mathcal{L}_{match} = \|\hat{Z}_{attr} W - \mathcal{E}_{struct}(A; \phi_S)\|_F^2$$

*Table 8.* Profiling of the attribute-level evaluation for node attribute reconstruction on the Amazon-Computer and Amazon-Photo dataset.

| Method | Amazon-Computer | | | | | | Amazon-Photo | | | | | |
|---|---|---|---|---|---|---|---|---|---|---|---|---|
| | R@10 | R@20 | R@50 | N@10 | N@20 | N@50 | R@10 | R@20 | R@50 | N@10 | N@20 | N@50 |
| VAE | 2.56 | 5.06 | 12.14 | 6.36 | 9.78 | 17.44 | 2.77 | 5.30 | 12.72 | 6.78 | 10.25 | 18.23 |
| GCN | 2.73 | 5.29 | 12.76 | 6.75 | 10.28 | 18.28 | 2.95 | 5.67 | 13.23 | 7.09 | 10.80 | 18.94 |
| GAT | 2.71 | 5.29 | 12.76 | 6.73 | 10.29 | 18.28 | 2.93 | 5.66 | 13.23 | 7.09 | 10.81 | 18.95 |
| NEIGHAGGR | 3.21 | 5.93 | 13.06 | 7.88 | 11.56 | 19.23 | 3.30 | 6.16 | 13.61 | 8.13 | 11.96 | 19.98 |
| GRAPHSAGE | 2.67 | 5.25 | 12.70 | 6.63 | 10.15 | 18.15 | 2.93 | 5.66 | 13.26 | 7.11 | 10.83 | 19.00 |
| GRAPHMAE | 1.30 | 2.60 | 6.51 | 4.57 | 7.09 | 12.99 | 1.34 | 2.68 | 6.71 | 4.68 | 7.25 | 13.29 |
| ARWMF | 2.82 | 5.40 | 12.81 | 6.91 | 10.48 | 18.32 | 2.99 | 5.71 | 13.25 | 7.24 | 10.89 | 19.09 |
| T2-GNN | 2.77 | 5.21 | 12.76 | 6.75 | 10.44 | 18.83 | 2.96 | 5.74 | 12.98 | 6.96 | 10.20 | 18.87 |
| GINN | 2.86 | 5.24 | 12.31 | 6.54 | 10.12 | 18.98 | 3.01 | 5.82 | 13.20 | 7.13 | 10.87 | 19.21 |
| SAT | 4.20 | 7.41 | 15.70 | 10.31 | 14.70 | 23.45 | 4.42 | 7.82 | 16.10 | 10.76 | 14.92 | 24.15 |
| AMER | 4.22 | 7.50 | 15.91 | 10.34 | 14.81 | 23.93 | 4.21 | 7.60 | 16.25 | 10.33 | 14.85 | 24.04 |
| WGNN | 4.45 | 7.76 | 16.22 | 10.90 | 15.32 | 24.26 | 4.37 | 7.74 | 16.30 | 10.66 | 15.21 | 24.29 |
| **GLAD** | 4.51 | 8.07 | 17.21 | 12.18 | 16.32 | 25.21 | 4.86 | 8.16 | 16.85 | 11.65 | 16.33 | 25.66 |

*Note:* The highest performance is shaded in green , and the second-highest performance is shaded in blue .

- **The Mathematical Flaw (Representation Collapse):** When backpropagating the error, the gradient w.r.t the GNN parameters $\nabla_{\phi_S}\mathcal{L}_{match}$ actively pulls the structure representation $Z_{struct}$ towards the generated (and initially noisy) $\hat{Z}_{attr}$. Because both distributions are dynamic ("moving targets"), the network can trivially minimize $\mathcal{L}_{match}$ by collapsing both $\hat{Z}_{attr}$ and $Z_{struct}$ into a constant, low-variance space ($\hat{Z}_{attr} \to \mathbf{c}, Z_{struct} \to \mathbf{c}$), leading to severe representation collapse.

- **Why GLAD's design is superior:** In our $\mathcal{L}_{align} = \|A - \mathcal{D}_{struct}(\hat{Z}_{attr})\|_F^2$, the target $A$ is a constant tensor ($\nabla_A\mathcal{L}_{align} = 0$). The model cannot "cheat" by altering the ground truth topology; it is strictly forced to learn semantically rich representations in $\hat{Z}_{attr}$ to reconstruct the exact 0/1 edges.

### *Alternative B: Joint Diffusion of Structure and Attributes*

Another alternative is to diffuse the topology and attributes simultaneously. Let the joint state be $\mathcal{Z}_t = [Z_{attr}^{(t)}, A^{(t)}]$. The reverse denoising network would model the joint transition $p_\theta(\mathcal{Z}_{t-1}|\mathcal{Z}_t)$.

- **The Mathematical Flaw (Modality Mismatch & $\mathcal{O}(N^2)$ Complexity):**
  1. *Modality Heterogeneity:* Standard Gaussian diffusion assumes continuous spaces $\mathcal{N}(\mu, \Sigma)$. Adding Gaussian noise to a discrete, highly sparse matrix $A \in \{0, 1\}^{N \times N}$ aggressively destroys higher-order topological properties (e.g., triangle closures, power-law degrees) much faster than it degrades continuous features $X$, making the joint score estimation highly unstable.
  2. *Complexity Explosion:* Modeling the full joint distribution across $T$ steps requires the network to predict the noise for the structural component at every timestep. Reconstructing the $N \times N$ adjacency matrix over $T$ steps yields an intractable time complexity of $\mathcal{O}(N^2 \cdot d \cdot T)$. For our large-world traffic graphs ($N \approx 60,000$), this is computationally impossible.

- **Why GLAD's design is superior:** GLAD performs the expensive, $T$-step iterative diffusion purely in the highly compressed, continuous $\mathcal{O}(N)$ attribute space. The bidirectional structural constraint is evaluated **only once** at $t = 0$. Furthermore, as detailed in our **Algorithm 4**, by reformulating $\mathcal{L}_{align}$ with negative sampling, we compute the loss only on the non-zero edges $\mathcal{E}^+$ and sampled negative edges $\mathcal{E}^-$:

$$\mathcal{L}_{align} \approx \sum_{(i,j)\in\mathcal{E}^+\cup\mathcal{E}^-} BCE\big(\sigma(\hat{z}_{0,i}^T\hat{z}_{0,j}), A_{ij}\big)$$

This brilliantly drops the complexity from $\mathcal{O}(N^2 \cdot T)$ to a highly scalable $\mathcal{O}(|E| + M)$, easily handling massive graphs.

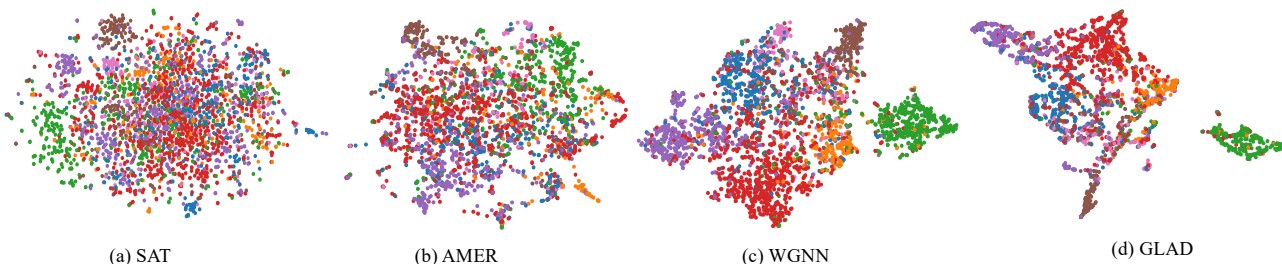

*Figure 9.* Radar chart in terms of {Recall@10, Recall@20, Recall@50, NDCG@10, NDCG@20, NDCG@50} .

*Figure 10.* Visualization of node representations generated by four methods on the Cora dataset.

*Table 9.* Hyperparameter analysis on Cora and London datasets. $\lambda$ (alignment weight), $w$ (guidance scale), and $T$ (diffusion steps).

| Hyperparameter | Value | Cora: Recall@10 ($\uparrow$) | Cora: Acc (%) ($\uparrow$) | London: MAE ($\downarrow$) |
|---|---|---|---|---|
| Alignment Loss Weight ($\lambda$) Default: $\lambda = 1.0$ | 0.1 | 11.24 | 81.3 | 88.50 |
| | 1.0 | **17.32** | **86.0** | **80.01** |
| | 5.0 | 15.65 | 84.2 | 83.15 |
| Guidance Scale ($w$) Default: $w = 0.5$ | 0.0 | 14.15 | 82.5 | 85.32 |
| | 0.5 | **17.32** | **86.0** | **80.01** |
| | 2.0 | 16.50 | 85.1 | 81.65 |
| Diffusion Steps ($T$) Default: $T = 100$ | 50 | 15.80 | 83.6 | 84.10 |
| | 100 | **17.32** | **86.0** | **80.01** |
| | 500 | **17.38** | **86.2** | 79.85 |

*Table 10.* Robustness to feature noise. We added zero-mean Gaussian noise to the observed attributes.

| Method | Cora: NDCG@50 ($\uparrow$) | | | London: MAPE (%) ($\downarrow$) | | |
|---|---|---|---|---|---|---|
| | 10% | 30% | 50% | 10% | 30% | 50% |
| SAT | 19.67 | 16.09 | 11.45 | 24.15 | 31.50 | 42.13 |
| GLAD (Ours) | **23.10** | **20.55** | **16.80** | **15.88** | **18.25** | **23.60** |

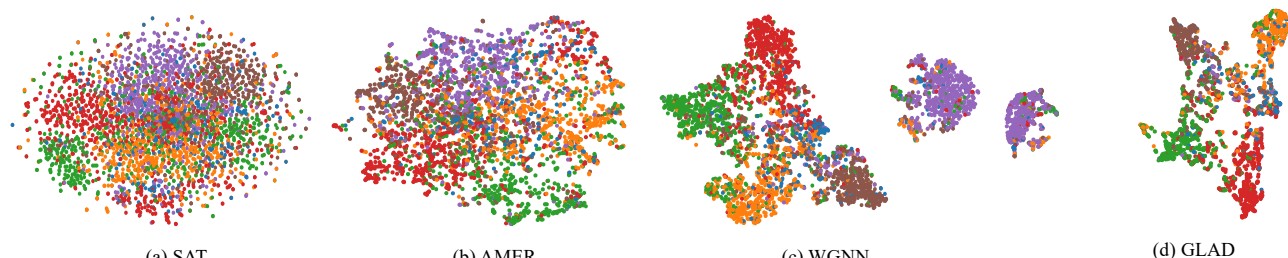

(a) SAT      (b) AMER      (c) WGNN      (d) GLAD

*Figure 11.* Visualization of node representations generated by four methods on the Citeseer dataset.

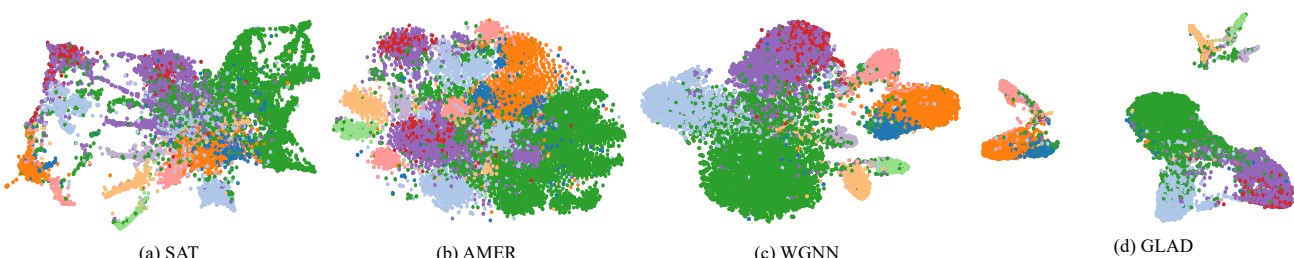

(a) SAT      (b) AMER      (c) WGNN      (d) GLAD

*Figure 12.* Visualization of node representations generated by four methods on the Amazon-Computer dataset.

