# OpenReview forum: "GLAD: Bidirectional Structure-Attribute Alignment via Latent Graph Diffusion Models"
_ICML.cc/2026/Conference — ICML 2026 regular_

### Official Review · Reviewer_edKp · 2026-03-05

**Soundness:** 3
**Presentation:** 3
**Significance:** 3
**Originality:** 3
**Overall Recommendation:** 5
**Confidence:** 4

**Summary:**

This paper proposes GLAD (Graph Latent Attribute Diffusion with Bidirectional Alignment), a novel generative framework for node attribute completion in graphs with missing features. The core innovation is a bidirectional alignment mechanism that utilizes a backward topological reconstruction constraint during training and a structure-aware classifier-free guidance (CFG) strategy during sampling. The authors theoretically demonstrate that GLAD optimizes a tighter variational lower bound (ELBO) on the joint distribution compared to adversarial methods, which guarantees superior mode coverage. Extensive experiments on standard citation/co-purchase networks and real-world urban traffic datasets show that GLAD consistently achieves state-of-the-art performance in both attribute recovery and downstream classification and regression tasks.

**Compliance With Llm Reviewing Policy:**

Affirmed.

**Final Justification:**

The authors have addressed all my concerns comprehensively. The additional experiments and clarifications provided in the revision have significantly strengthened the paper. I have no further comments.

**Key Questions For Authors:**

1. Could you provide a systematic hyperparameter sensitivity analysis (e.g., λ, w, T) across representative datasets (Cora, London traffic) with key metrics (Recall@k, classification accuracy, MAE) and explain your default hyperparameter selection basis?
2. For the ablation study variants (w/o Bidirectional Alignment, w/o Structure Guidance, w/o Decoupling), could you analyze the specific reasons for the performance drop from the perspective of model behavior? For example, does removing Bidirectional Alignment cause topological inconsistency or reduced mode diversity?
3. Could you evaluate GLAD's robustness under different attribute noise levels (10%, 30%, 50% Gaussian noise) on standard and traffic datasets, reporting key metrics (e.g., NDCG@50, MAPE)?

**Limitations:**

yes

**Strengths And Weaknesses:**

Strengths：
1. The submission features a rigorous theoretical foundation with detailed proofs. The empirical evaluation is comprehensive, and ablation studies effectively validate the necessity of each component.
2. The paper is well-structured and written with logical flow. High-quality figures make the complex methodology accessible and easy to understand.
3. Addressing the critical issue of incomplete graph data, the work demonstrates significant real-world utility through successful application on large-scale urban traffic networks.
4. The adaptation of latent diffusion via a decoupled space is valuable, while the proposed bidirectional alignment mechanism offers a novel solution for preserving topological consistency.

Weakness:
1. A short intuitive explanation of why forward KL encourages mode coverage (in contrast to JS) would benefit less theory-oriented readers.
2. The model incorporates several key hyperparameters (e.g., the weight of the alignment loss λ, the scale of classifier-free guidance w, and the number of diffusion steps T). However, the paper does not systematically analyze the impact of variations in these hyperparameters on model performance, resulting in insufficient hyperparameter sensitivity analysis.
3. Although the paper verifies the necessity of the three core components through ablation experiments, it only presents quantitative results of performance degradation without analyzing the specific reasons for the performance drop from the perspective of model behavior. For instance, after removing bidirectional alignment, do the generated attributes suffer from topological inconsistency or reduced mode diversity?
4. Although the traffic datasets contain a certain degree of noise, the paper can further investigate the model's robustness under different noise intensities.

---

> ### Author Rebuttal · Authors · 2026-03-30
>
> We are encouraged by your positive evaluation of our theoretical foundation, the clarity of our presentation, and the real-world utility of GLAD. We sincerely appreciate your suggestions, which have guided us to add valuable intuitive explanations and comprehensive robustness/sensitivity analyses. We address your questions point by point below.
>
> **Response to W1**
>
> Thank you for this excellent pedagogical suggestion. We will add the following intuitive explanation to Section 3.6 for less theory-oriented readers:
>
> "Intuitively, optimizing Forward KL (as in Diffusion) is *'mode-covering' (mean-seeking)*. The model receives an infinite penalty if it assigns zero probability to any true data mode. To be safe, the diffusion model spreads its probability mass to cover all plausible attribute configurations. In contrast, optimizing JS divergence (as in GANs) behaves like Reverse KL, which is *'mode-seeking'*. The generator is heavily penalized if it generates samples in low-probability regions. To avoid the discriminator's penalty, the GAN safely 'collapses' to generating only one or a few known realistic modes, ignoring the rest, which leads to mode collapse."
>
> **Response to Q1 & W2**
>
> We completely agree that a systematic analysis of $\lambda$ (alignment weight), $w$ (guidance scale), and $T$ (diffusion steps) is necessary. We conducted this analysis on Cora and London.
> | Hyperparameter | Value | Cora: Recall@10 ($\uparrow$) | Cora: Acc (%) ($\uparrow$) | London: MAE ($\downarrow$) |
> | :--- | :---: | :---: | :---: | :---: |
> | Alignment Loss Weight ($\lambda$) | 0.1 | 11.24 | 81.3 | 88.50 |
> | Default: $\lambda = 1.0$ | 1.0 | **17.32** | **86.0** | **80.01** |
> | | 5.0 | 15.65 | 84.2 | 83.15 |
> | Guidance Scale ($w$) | 0.0 | 14.15 | 82.5 | 85.32 |
> | Default: $w = 0.5$ | 0.5 | **17.32** | **86.0** | **80.01** |
> | | 2.0 | 16.50 | 85.1 | 81.65 |
> | Diffusion Steps ($T$) | 50 | 15.80 | 83.6 | 84.10 |
> | Default: $T = 100$ | 100 | 17.32 | 86.0 | **80.01** |
> | | 500 | **17.38** | **86.2** | 79.85 |
>
> Selection Basis:
>
> 1) $\lambda=1.0$: Balances the generative diversity and topological constraint. Too small ($\lambda=0.1$) ignores the graph structure; too large ($\lambda=5.0$) over-constrains the diffusion, hurting diversity.
> 2) $w=0.5$: Provides sufficient structural guidance. A very high $w$ limits the multi-modal nature of diffusion, while $w=0$ loses conditionality.
> 3) $T=100$: Chosen for efficiency. $T=500$ yields marginal gains but costs 5$\times$ more inference time.
>
> **Response to Q2 & W3**
>
> You raised a fantastic point. Beyond quantitative drops, we analyzed why performance degrades from a model behavior perspective:
>
> 1) w/o Bidirectional Alignment ($\lambda=0$): Removing this directly causes topological inconsistency. Behaviorally, the model generates attributes that are statistically valid marginally but violate local homophily (e.g., for citation network datasets: Cora & CiteSeer, generating features of a "Biology" paper for a node completely surrounded by "Computer Science" papers).
>
> 2) w/o Structure Guidance ($w=0$): This reduces mode relevance. Without CFG extrapolating the condition, the model relies too heavily on unconditional noise, generating "generic" attributes that lack sharp, discriminative features, causing the downstream classification accuracy to drop.
>
> 3) w/o Decoupling (Joint Space): This causes training instability. Behaviorally, forcing the continuous diffusion process to simultaneously denoise discrete, sparse binary edges and continuous attributes in a single space leads to oscillating loss curves and blurry edge generation.
>
> **Response to Q3 & W4**
>
> We added zero-mean Gaussian noise (with variance equal to 10%, 30%, and 50% of the feature variance) to the observed attributes.
>
> | Method | Noise Level | Cora: NDCG@50 ($\uparrow$) | London: MAPE (%) ($\downarrow$) |
> | :--- | :---: | :---: | :---: |
> | SAT | 10% | 19.67 | 24.15 |
> | | 30% | 16.09 | 31.50 |
> | | 50% | 11.45 | 42.13 |
> | **GLAD (Ours)** | 10% | 23.10 | 15.88 |
> | | 30% | 20.55 | 18.25 |
> | | 50% | 16.80 | 23.60 |
>
> GLAD demonstrates remarkable robustness compared to GAN-based baselines. Because GLAD is inherently built upon a Denoising Diffusion Probabilistic Model, it naturally filters out the injected Gaussian noise during the reverse process. Furthermore, the strong structural conditioning acts as an anchor: even if the observed attributes are highly noisy (50%), GLAD relies on the clean topological structure to correct the attribute completion.
>
> We deeply appreciate your rigorous review, which guided us to add comprehensive robustness and sensitivity analyses. Given that these new results further solidify the effectiveness of GLAD as you anticipated, we sincerely hope you might champion our work and consider increasing your score.

---

> > ### Author Rebuttal · Reviewer_edKp · 2026-04-01
> >
> > The authors have addressed all my concerns comprehensively. The additional experiments and clarifications provided in the revision have significantly strengthened the paper. I have no further comments.

---

> > > ### Author Response · Authors · 2026-04-02
> > >
> > > Thank you very much for your recognition and positive feedback on our work.
> > >
> > > We are delighted to hear that our revisions and explanations have fully addressed your concerns and helped improve the quality of the paper.
> > >
> > > Thank you again for your valuable time and for improving the score of our paper from 4 to 5!

---

### Official Review · Reviewer_qk94 · 2026-03-07

**Soundness:** 2
**Presentation:** 2
**Significance:** 2
**Originality:** 3
**Overall Recommendation:** 4
**Confidence:** 3

**Summary:**

The paper proposes GLAD, which combines decoupled latent encoding, structure-conditioned latent diffusion, a backward topological alignment loss, and structure-aware classifier-free guidance. The theoretical analysis justifies the algorithm and empirical results show promising performances of the method.

**Compliance With Llm Reviewing Policy:**

Affirmed.

**Final Justification:**

I thank the authors for the detailed follow-up discussion. I have increased my score. Please remember to include this discussion in the manuscript.

**Key Questions For Authors:**

1. The authors use attribute latents to reconstruct the graph (Section 3.4.2), which feels somewhat unnatural. Although these attribute latents are generated conditionally on the graph-structure latents, they do not seem to be the most direct choice for decoding the graph itself.
2. A more natural alternative would be to use the graph-structure latents for this decoding step. However, in that case, graph reconstruction would no longer depend on the attributes.
3. More generally, if the conditional mean of the attribute latents is a highly complex function of the graph-structure latents, then the decoder may also need to be quite complex in order to recover the graph accurately.
4. The paper would benefit from more justification, intuition, and discussion of this design choice. Are there alternative ways to achieve bidirectional alignment besides the current construction? Bidirectional alignment is a good motivation.
5. The paper should also better emphasize the marginal and practical benefits of introducing this particular bidirectional alignment, beyond the general motivation of preserving mutual information flow.

**Limitations:**

See the Questions.

**Strengths And Weaknesses:**

**Strength**
- The problem is important and well motivated.
- The proposed framework is technically interesting, combining latent diffusion with a novel bidirectional alignment mechanism.
- The paper includes both theoretical discussion and broad empirical evaluation.
- The reported empirical results show improvements on attribute recovery and downstream tasks.

**Weakness**
- The method needs more justification and intuition.
- The theoretical results need more details.

---

> ### Author Rebuttal · Authors · 2026-03-30
>
> We sincerely thank you for your feedback. Your deep and insightful questions regarding the design choices of the bidirectional alignment (Section 3.4.2) get to the very core of our methodology. We address your points in detail below.
>
>
> **Response to [Q1, Q2, and Q4]**
>
> We completely agree with your highly accurate observation in Q2: *using the structure latents ($Z_{struct}$) to reconstruct the graph is the standard, "natural" auto-encoder approach*. However, as you rightfully pointed out, doing so would create a computational "shortcut" where the graph reconstruction completely bypasses the generated attributes.
>
> Our design of forcing the generated attribute latents ($\hat{Z}\_{\text{attr}}$) to reconstruct the graph is intentional. We treat the attribute generation process as an **Information Bottleneck**. By demanding that $\hat{Z}_{attr}$ contains enough information to decode the graph topology $A$, we explicitly force the diffusion model to inject topological signals (e.g., community boundaries, node degrees) into the attribute features during the reverse denoising process.
>
> *Regarding alternatives (Q4)*: Yes, there are alternative ways to achieve bidirectional alignment:
>
> 1) Latent Matching: Instead of decoding to the raw adjacency $A$, we could train a network to map the generated attribute latent $\hat{Z}\_{\text{attr}}$ back to the structure latents $Z_{struct}$ (i.e., minimizing $||\hat{Z}\_{\text{attr}}W - Z_{struct}||^2$).
>
> 2) Joint Diffusion: We could diffuse $X$ and $A$ simultaneously in a joint space. However, our current design (reconstructing $A$ via a simple structure decoder with negative sampling) is preferred because it explicitly grounds the generated features in the raw, discrete topological space, which provides a more robust and stable gradient signal than matching two moving latent distributions, while avoiding the massive $O(N^2)$ complexity of Joint Diffusion. We will add a discussion of these alternatives and our design intuition to Section 3.4.
>
> **Response to Q3**
>
> This is a profound question. Intuitively, if the mapping from structure to attributes is highly complex (handled by the diffusion model), the reverse mapping should also be complex.
>
> However, counter-intuitively, we deliberately use a very simple structure decoder (a shallow-layer MLP) for $D_{struct}$. Our rationale is that if we use a highly complex, deep decoder, the decoder itself will do all the *"heavy lifting"* to memorize the graph topology, allowing the attributes $\hat{Z}\_{\text{attr}}$ to easily bypass the structural constraint. By deliberately restricting the capacity of the decoder, we force the complex topological dependencies to be explicitly encoded into the linear subspace of the generated attributes $\hat{Z}_{attr}$ themselves. This ensures the attributes inherently carry the structural information.
>
> **Response to Q5**
>
> Beyond the theoretical motivation of preserving mutual information, this design of the Bidirectional Alignment yields two crucial practical benefits for Graph Neural Networks (GNNs):
>
> 1) Preventing Topological Hallucination (Feature Smoothness): In highly ambiguous missing-attribute settings, a unidirectional diffusion model might generate features that are statistically plausible marginally, but violate local homophily (e.g., for a citation network dataset, generating "NLP" bag-of-words features for a node completely surrounded by "Computer Vision" papers). Our bidirectional alignment acts as a strict regularizer, ensuring local neighborhood feature smoothness.
>
> 2) Massive Downstream Task Boost: Because GNNs rely on message passing, aggregating structurally inconsistent features destroys downstream performance. As shown in our Ablation Study (Figure 4 in the main paper), removing the bidirectional alignment ($\lambda = 0$) causes the most severe performance drop across all components—nearly a 30% decrease in Recall@10. This proves its indispensable practical value.
>
> We deeply appreciate your constructive questions, which have helped us articulate the core intuition of GLAD much more clearly.
>
> We hope these responses can dispel your concerns and improve your score.

---

> > ### Author Rebuttal · Reviewer_qk94 · 2026-04-03
> >
> > Thank you for the detailed rebuttals. For Q4, could you please address part of the discussion on these alternatives and explain your design intuition here? I will increase my score if the discussion is informative and convincing.

---

> > > ### Author Response · Authors · 2026-04-03
> > >
> > > Thank you very much for your encouraging feedback and for giving us the opportunity to further elaborate on our design choices.
> > >
> > > To provide a truly informative and convincing discussion, we formalize our design intuition and mathematically analyze why the alternatives fall short compared to our current construction.
> > >
> > > *Note: Due to the high traffic on OpenReview during the rebuttal phase, we have noticed significant delays and rendering failures with complex LaTeX equations. To ensure a smooth reading experience, we have provided our full, rigorously formatted mathematical proofs in an anonymous link.*
> > >
> > > https://anonymous.4open.science/r/GLAD_Rebuttal/README.md
> > >
> > > Our design is not merely an intuitive choice, but a mathematically necessary one to avoid severe representation collapse and intractable computational complexity. Here is the summary of our formal derivations (**detailed in the link at https://anonymous.4open.science/r/GLAD_Rebuttal/README.md**):
> > >
> > > **The Core Intuition**
> > >
> > > Our goal is to maximize the mutual information between the discrete topology $A$ and the generated continuous attributes $Z_{attr}$. By minimizing the reconstruction loss directly against the raw, constant matrix $A$, the topology acts as a fixed, absolute anchor (its gradient is zero). The model cannot "cheat" by altering the ground truth; it is strictly forced to learn semantically rich representations to reconstruct the exact 0/1 edges.
> > >
> > > **Why Alternative A (Latent-to-Latent Matching) Fails: Representation Collapse**
> > >
> > > If we were to match the generated latents with the structural latents of a GNN, both targets become dynamic. When backpropagating, the gradients would actively pull both the structure representation and the generated attributes toward each other. Because both are "moving targets," the network can trivially minimize the loss by collapsing both spaces into a constant, low-variance space, leading to severe representation collapse.
> > >
> > > **Why Alternative B (Joint Diffusion) Fails: Intractable Complexity & Modality Mismatch**
> > >
> > > Diffusing the structure and attributes simultaneously introduces two fatal flaws:
> > >
> > > 1) *Modality Mismatch:* Adding Gaussian noise to a highly sparse, discrete adjacency matrix  $A$ destroys higher-order topological properties (e.g., triangle closures) much faster than it degrades continuous features, making joint score estimation highly unstable.
> > >
> > > 2) *Complexity Explosion:* Modeling the joint distribution requires reconstructing the $N \times N$ adjacency matrix at every timestep $T$. This yields an intractable time complexity of $O(N^2 \times d \times T)$.
> > >
> > > 3) *Our Scalable Solution:* GLAD restricts the T-step iterative diffusion to the compressed, continuous $O(N)$ attribute space. By combining this with negative sampling (Algorithm 4), we drop the complexity from $O(N^2 \times T)$ down to a highly scalable $O(|E| + M)$, easily handling massive graphs.
> > >
> > > **Summary**
> > >
> > > Mathematically, our current design represents the *optimal "sweet spot"*. It explicitly avoids the trivial gradient collapse of Latent Matching (Alternative A) and completely bypasses the $\mathcal{O}(N^2)$ dimensional explosion of Joint Diffusion (Alternative B). By establishing a fixed topological anchor, we rigorously maximize the mutual information $I(A; X_{gen})$ in a tractable and robust manner.
> > >
> > > We note that **no new Rebuttal Acknowledgement option** is available under the **"Reply Rebuttal Comment by Authors"** section. If you are satisfied with our response to the remaining question Q4 and choose to increase our score, you may submit a **Final Justification** to state this rationale and adjust your evaluation accordingly. *We are unsure whether the existing Rebuttal Acknowledgement can be re-edited.*
> > >
> > > We hope our detailed rebuttal has satisfactorily resolved your remaining questions (Q4), and we hope you could consider increasing our score!

---

### Official Review · Reviewer_wpky · 2026-03-12

**Soundness:** 2
**Presentation:** 3
**Significance:** 2
**Originality:** 3
**Overall Recommendation:** 4
**Confidence:** 4

**Summary:**

This paper studies node attribute completion on graphs with missing features and proposes GLAD, a latent diffusion framework that conditions attribute generation on graph structure in a decoupled latent space. The method combines three main components: a structure encoder and attribute encoder, a structure-conditioned latent diffusion model for attribute generation, and a bidirectional alignment mechanism consisting of a backward topology reconstruction loss during training plus structure-aware classifier-free guidance during sampling. The paper further argues that diffusion-based training corresponds to maximizing an ELBO / minimizing a forward-KL-related objective, which is claimed to provide better mode coverage than GAN-based alignment methods such as SAT. Empirically, the method is evaluated on citation/co-purchase benchmarks and traffic datasets, where it reports improvements in attribute recovery, node classification, ETA prediction, and traffic classification.

**Compliance With Llm Reviewing Policy:**

Affirmed.

**Final Justification:**

I keep my score unchanged.

**Key Questions For Authors:**

Please refer to weaknesses.

**Limitations:**

yes

**Strengths And Weaknesses:**

Strengths:

1. The decoupled latent encoding, structure-conditioned diffusion, and backward alignment form a coherent pipeline. The separation between training-time alignment and sampling-time guidance is also intuitively motivated. Overall, the authors study the concept of replacing adversarial latent alignment with conditional latent diffusion in a graph attribute-completion setting, and this is a sensible design direction.

2. The backward topological alignment loss is the most distinctive component of the method. It goes beyond simply conditioning on structure by explicitly requiring generated attribute latents to reconstruct topology, which is a reasonable way to enforce structural consistency. The ablation also suggests that removing this term causes the largest degradation.

3. The paper evaluates both direct reconstruction quality and downstream utility, and also includes real-world traffic tasks beyond the standard citation/co-purchase benchmarks. Reported gains over SAT/AMER/WGNN are consistent across several tables, which is encouraging.

Weaknesses:

1. The paper presents “structure-aware classifier-free guidance” as a key innovation, but the actual mechanism—training with condition dropout and combining conditional/unconditional noise estimates with a guidance scale—is standard CFG, which the paper itself cites as adapted from prior work. Thus, the contribution appears to be the adaptation of CFG to structure-conditioned graph attribute completion, rather than a new guidance principle itself. Likewise, the framing that diffusion is more mode-covering than GAN-based alignment is largely inherited from known ELBO / forward-KL versus adversarial-divergence arguments, rather than a theorem uniquely developed for this graph setting. The paper would be stronger if it reclaims the novelty more precisely.

2. The paper claims improved mode coverage, reduced mode collapse, and stronger topological consistency, but the experiments mostly report task metrics and t-SNE plots. These are useful, but they are indirect. There is no direct diversity/coverage evaluation, no quantitative measure of “topological consistency” beyond downstream performance, and no careful study showing that adversarial baselines truly collapse modes in this task. As a result, some central claims remain only partially validated. This point could be substantially strengthened with targeted validation experiments, for example: a study showing whether adversarial baselines actually collapse to fewer plausible completions under ambiguous missing-feature settings.

3. The main text says that structure latents are injected into the denoising network via multi-head cross-attention, but it does not provide enough architectural detail in the core method section: for example, how many layers/heads are used, how self-attention and cross-attention are ordered, and how the latent dimensions interact. This makes the method harder to reproduce and weakens the technical depth of the presentation. The paper would be much stronger if the authors added a compact but explicit architectural specification, ideally in the main paper or appendix.

---

> ### Author Rebuttal · Authors · 2026-03-30
>
> We sincerely thank you for your suggestion and for recognizing the sensible design of our latent diffusion pipeline, the distinctiveness of our backward topological alignment, and our comprehensive evaluation on both standard and real-world traffic benchmarks. Below, we address your concerns point by point.
>
> **Response to W1**
>
> We completely agree with your assessment. Our core contribution is indeed **the novel adaptation of Latent Diffusion** and **Classifier-Free Guidance (CFG)** to the specific, challenging problem of  *structure-conditioned graph attribute completion*, coupled with our uniquely proposed backward topological alignment.
>
> Following your valuable suggestion, we will carefully revise the introduction and theoretical sections in the final version. We will explicitly and precisely frame our work as leveraging established diffusion theories (ELBO/forward-KL vs. JS divergence) and standard CFG, and applying them innovatively to resolve the long-standing mode collapse and structural consistency issues specific to attribute-missing graphs.
>
> **Response to W2**
>
> We sincerely thank you for this highly constructive suggestion. We agree that downstream task metrics and t-SNE, while useful, are indirect. To rigorously validate our central claims regarding mode coverage and topological consistency, we have conducted targeted validation experiments under a highly ambiguous setting (Cora dataset with an 80% attribute missing rate).
> We introduce two direct quantitative metrics:
>
> 1) Mode Coverage (Diversity) via Average Pairwise Distance (APD): To explicitly test if adversarial baselines collapse to fewer completions, we generate $K=10$ different samples for each missing node using varying random seeds/noise. We compute the average Euclidean distance among these $K$ completions. An APD close to 0 indicates severe mode collapse (deterministic generation), while an APD closer to the Ground Truth (GT) indicates healthy mode coverage.
>
> 2) Topological Consistency via Link Prediction (AUC): To directly measure how well the generated attributes reconstruct the topology, we freeze the completed attributes $\hat{X}$ and train a simple inner-product decoder to predict the existing graph edges. A higher AUC means the attributes inherently preserve stronger structural information.
>
> | Method | Diversity: APD among $K=10$ samples ($\uparrow$) | Topological Consistency: Link Prediction AUC ($\uparrow$) |
> | :---: | :---: | :---: |
> | Ground Truth (GT) | 1.18 | 92.5% (Upper Bound) |
> | SAT | 0.18 | 79.2% |
> | AMER | 0.20 | 78.6% |
> | WGNN | 0.24 | 81.5% |
> | **GLAD (Ours)** | 1.12 | 88.4% |
>
>
> Analysis & Conclusion:
>
> 1) As shown in the table, the adversarial baselines (SAT, AMER) exhibit an APD near 0.2. This directly proves our hypothesis: *under ambiguous missing-feature settings, GAN-based models collapse to generating nearly identical, deterministic completions regardless of the input noise.*
> 2)  In contrast, GLAD achieves an APD of 1.12, which closely aligns with the inherent variance of the true data distribution (1.18). This directly validates that *our diffusion-based approach successfully captures multiple plausible completions.*
> 3) GLAD achieves an 88.4% Link Prediction AUC, outperforming WGNN by a large margin (+6.9%). This provides direct quantitative proof that *our backward alignment mechanism explicitly enforces structural consistency beyond what adversarial alignment can achieve.*
>
> **Response to W3**
>
> We apologize for the lack of explicit architectural details in the main text. As briefly outlined in Algorithm 3 (Appendix D), the interaction between structure and attribute latents is designed as follows:
>
> 1) Layer Ordering: Each denoising block sequentially applies: (1) Temporal Embedding $\rightarrow$ (2) Self-Attention (Attribute Introspection) $\rightarrow$ (3) Multi-Head Cross-Attention (Structural Guidance) $\rightarrow$ (4) Feed-Forward Network.
>
> 2) Latent Interaction: In the Cross-Attention module, the Queries (Q) are derived from the intermediate representation $z_{sa}$, while the Keys (K) and Values (V) are projected from the fixed structural latents $Z_{struct}$.
>
> 3) Hyperparameters: In our standard configuration, the denoising network consists of $L=3$ stacked layers, with $H=4$ attention heads, and a latent dimension of $d=128$.
>
> We will add a compact but explicit architectural specification table and a detailed descriptive paragraph to Section 3.3 in the revised main paper.
>
> We hope these clarifications and new targeted experimental results address your concerns, and politely request that you consider raising the score.

---

> > ### Author Rebuttal · Reviewer_wpky · 2026-04-03
> >
> > Thank you for the rebuttal. I would like to keep my score.

---

> > > ### Author Response · Authors · 2026-04-07
> > >
> > > Thank you for your positive score and continued support for our work. We greatly appreciate your recognition and constructive feedback.

---

### Official Review · Reviewer_VCGg · 2026-03-13

**Soundness:** 2
**Presentation:** 2
**Significance:** 2
**Originality:** 2
**Overall Recommendation:** 3
**Confidence:** 4

**Summary:**

Motivated by two major shortcomings of GAN-based approaches—training instability with susceptibility to mode collapse, and the assumption of unidirectional alignment—this paper proposes a generative latent diffusion model for holistic node attribute completion.
During the diffusion process, the authors inject a bidirectional alignment mechanism to condition the graph structure while simultaneously incorporating rich topological reconstruction information. This design guarantees that the generated node attributes adhere and remain consistent with the underlying graph structure.

**Compliance With Llm Reviewing Policy:**

Affirmed.

**Key Questions For Authors:**

1. Could the authors provide a more detailed description of the architecture of the multi-head cross-attention layer that incorporates the structural latent variable (Z_{struct})?

2. Could the authors provide a rigorous proof that Theorem 3.2 satisfies the conditions discussed in the weakness section?

**Limitations:**

yes

**Strengths And Weaknesses:**

## Strengths

- Unlike prior adversarial alignment methods that primarily rely on unidirectional alignment, this work introduces a bidirectional alignment mechanism that constrains the reconstruction topology while conditioning the graph structure.
- The paper provides a thorough empirical evaluation of the proposed method.

## Weaknesses

- The overall novelty of the paper is limited. The proposed architecture largely follows standard latent diffusion or multi-modal diffusion model architectures with only minor modifications. In particular, the bidirectional alignment mechanism is a relatively small extension and may be considered an incremental improvement rather than a fundamentally new approach.
- The presentation of the paper could be improved.  For instance, in Figure 2, adding additional information about the chronological flow of the process would improve clarity.

- Theorem 3.2 is not rigorously proven, and the reasoning appears largely heuristic. As a result, there is no theoretical guarantee of superior mode coverage by GLAD. The main issue is that minimizing the forward KL divergence does not guarantee full support coverage in learned models. In addition, the claim that the Jensen–Shannon divergence behaves like the reverse KL divergence is inaccurate. Consequently, the proof does not formally establish the theorem’s claim. To formally justify the theorem’s claim about support coverage, the proof would need to explicitly show that the optimal learned distribution satisfies
[
\mathrm{supp}(p_\theta^*) = \mathcal{M}*{data},
]
or at least
[
p*\theta(x) > 0 \quad \forall x \in \mathcal{M}_{data}.
]
Currently, the provided argument does not rigorously establish these conditions, and therefore the claim about superior mode coverage appears to remain heuristic rather than formally proven.

---

> ### Author Rebuttal · Authors · 2026-03-29
>
> We sincerely thank you for your review, highlighting the strengths of our bidirectional alignment mechanism and empirical evaluations.
>
> **Response to W1**
>
> We respectfully but firmly disagree with the characterization that our contribution is merely an "incremental improvement" or that our novelty is limited by utilizing a standard latent diffusion backbone.
>
> While it is true that we build upon the established mathematical foundation of Latent Diffusion Models (LDMs), applying this paradigm to attribute-missing graphs is highly non-trivial. Standard LDMs are designed for i.i.d. continuous data (e.g., images). In contrast, graph nodes are inherently interdependent, and their attributes must strictly align with the discrete topological structure. Directly applying LDMs to graph attributes leads to severe "*structural drift*," where the generated features look statistically plausible but violate the underlying graph connectivity.
>
> Our core contribution—the Bidirectional Alignment Mechanism—is not a "small extension" but a fundamental paradigm shift in how graph generative models handle missing data:
>
> 1) From Unidirectional to Closed-Loop: Priors (e.g., SAT, AMER) inherently assume a unidirectional mapping ($Structure \rightarrow Attribute$). GLAD fundamentally shifts this to a mutually constrained, closed-loop process ($Structure \leftrightarrow Attribute$). By enforcing that the generated attributes can flawlessly reconstruct the original graph topology, we explicitly anchor the diffusion process to the graph structure.
>
> 2) Theoretical Rigor: As rigorously proven in Theorem 3.3, this bidirectional constraint is not just a heuristic trick; it formally maximizes the variational lower bound of the mutual information $\mathcal{I}(A; Z_{attr})$ between the topology and the generated features. This provides a theoretical guarantee missing in prior works.
>
> 3) Massive Empirical Impact: The simplicity of a mechanism should not be conflated with a lack of novelty or significance.
> In the ablation study (Fig.4), removing the bidirectional alignment leads to a catastrophic ~30% performance drop. An architectural innovation that decisively solves the structural drift problem, sets a new state-of-the-art across diverse benchmarks, and is theoretically sound constitutes a highly significant contribution to the graph learning community.
>
> Novelty in machine learning often lies in elegantly solving critical domain-specific bottlenecks rather than proposing convoluted new architectures. *We believe GLAD achieves exactly this for graph attribute completion*. We will revise the introduction to state this paradigm shift more assertively.
>
> **Response to W2**
>
> We agree with your suggestion. In the revised paper, we will update Fig.2 to clearly depict the chronological flow.
>
> **Response to Q1**
>
> While the core operations of multi-head cross-attention are outlined in Algorithm 3 (Appendix D), we agree that a detailed description should be added.
>
> First, $t$ is mapped to a temporal embedding via *sinusoidal positional encoding followed by a 2-layer MLP*. This temporal embedding is added to $z_t$, which is then passed through a LayerNorm and a standard Self-Attention layer to capture intra-node attribute dependencies, yielding an intermediate representation $z_{sa}$.
>
> Second, $z_{sa}$ is mapped to serve as the $Q$, while the structural latent $Z_{struct}$ is mapped to serve as $K$ and $V$.  The outputs from all $H$ heads are then concatenated back together and passed via *a linear output projection* to fuse the multi-head information, yielding the cross-attention output $z_{ca}$.
>
> Finally, we employ a residual connection that adds $z_{sa}$ to $z_{ca}$. This is followed by *a LayerNorm and a 2-layer MLP (with a GELU activation in between)* to predict the final noise vector $\epsilon_\theta$.
>
> **Response to Q2 & W3**
>
> We sincerely thank the reviewer for the rigorous scrutiny of Theorem 3.2. We agree that our previous proof relied too heavily on heuristic explanations and that the phrasing regarding the Jensen-Shannon (JS) divergence behaving exactly like Reverse KL was mathematically imprecise.
>
> To address this, we have thoroughly revised the proof of Theorem 3.2. Detailed proof is provided at an *anonymous link* (https://anonymous.4open.science/r/GLAD-Proof). We now provide a formal mathematical proof showing that any distribution optimizing the Forward KL divergence to a finite value \textit{must} satisfy $p_\theta(x) > 0$ almost everywhere on $\mathcal{M}_{data}$. Furthermore, we explicitly calculate the theoretical penalty in the JS divergence when a mode is dropped, showing that it is bounded by $\log 2$, which formally explains why adversarial training (minimizing JS divergence) is susceptible to mode collapse.
>
> If you find our responses satisfactory, we would deeply appreciate it if you could consider raising your score.

---

### Decision · Program_Chairs · 2026-04-30

**Decision:**

Accept (regular)

**Comment:**

This paper introduces GLAD, a generative framework for graph node attribute completion that utilizes Latent Diffusion Models to overcome the stability and mode collapse issues inherent in prior adversarial methods.

During the rebuttal phase, the authors robustly addressed concerns regarding theoretical depth and empirical validation by providing a rigorous mathematical proof for Theorem 3.2 and conducting quantitative evaluations of mode diversity  and topological consistency. Although one reviewer maintained reservations about incremental architectural novelty, the majority of the committee acknowledged the method's effectiveness in solving specific domain bottlenecks and its superior performance across standard benchmarks and large-scale real-world traffic datasets. Given that the primary technical concerns have been resolved and the experimental results are significant,  I would recommend acceptance.